# Crotamiton derivative JM03 extends lifespan and improves oxidative and hypertonic stress resistance in *Caenorhabditis elegans* via inhibiting OSM-9

**Keting Bao[1†], Wenwen Liu[2†], Zhouzhi Song[1†], Jiali Feng[1], Zhifan Mao[1], Lingyuan Bao[1], Tianyue Sun[1], Zelan Hu[1]\*, Jian Li[1,2,3,4]\***

[1]State Key Laboratory of Bioreactor Engineering, Shanghai Frontiers Science Center of Optogenetic Techniques for Cell Metabolism, Frontiers Science Center for Materiobiology and Dynamic Chemistry, Shanghai Key Laboratory of New Drug Design, School of Pharmacy, East China University of Science and Technology, Shanghai, China; [2]Key Laboratory of Tropical Biological Resources of Ministry of Education, College of Pharmacy, Hainan University, Haikou, China; [3]Yunnan Key Laboratory of Screening and Research on Anti-pathogenic Plant Resources from West Yunnan, College of Pharmacy, Dali University, Dali, China; [4]Clinical Medicine Scientific and Technical Innovation Center, Shanghai Tenth People's Hospital, Tongji University School of Medicine, Shanghai, China

**\*For correspondence:**
huzelan@ecust.edu.cn (ZH);
jianli@ecust.edu.cn (JL)

†These authors contributed equally to this work

**Competing interest:** The authors declare that no competing interests exist.

**Abstract** While screening our in-house 1072 marketed drugs for their ability to extend the lifespan using *Caenorhabditis elegans* (*C. elegans*) as an animal model, crotamiton (*N*-ethyl-o-crotonotoluidide) showed anti-aging activity and was selected for further structural optimization. After replacing the ortho-methyl of crotamiton with ortho-fluoro, crotamiton derivative JM03 was obtained and showed better activity in terms of lifespan-extension and stress resistance than crotamiton. It was further explored that JM03 extended the lifespan of *C. elegans* through osmotic avoidance abnormal-9 (OSM-9). Besides, JM03 improves the ability of nematode to resist oxidative stress and hypertonic stress through OSM-9, but not osm-9/capsaicin receptor related-2 (OCR-2). Then the inhibition of OSM-9 by JM03 reduces the aggregation of Q35 in *C. elegans* via upregulating the genes associated with proteostasis. SKN-1 signaling was also found to be activated after JM03 treatment, which might contribute to proteostasis, stress resistance and lifespan extension. In summary, this study explored a new small molecule derived from crotamiton, which has efficient anti-oxidative, anti-hypertonic, and anti-aging effects, and could further lead to promising application prospects.

## Editor's evaluation

The study proceeds from a large initial *C. elegans* lifespan screen, comprising over 1000 candidate drugs, to a successful application of structure-activity analysis to yield an optimized lead compound and a likely mechanistic target. In addition to reporting novel compounds affecting lifespan, this work will be of interest to other researchers working at different stages of drug discovery, repurposing and lead optimization.

## Introduction

In spite the fact that aging is an inevitable process, many efforts have been made to uncover drugs which could delay aging. Numerous aging associated signaling pathways, discovered in *Caenorhabditis elegans*, are found to be conserved in the mammals (*Kenyon, 2010*). Moreover, many compounds, which extended the lifespan of *C. elegans*, also showed anti-aging effects in the mice model. For example, urolithin A was found to prolong the lifespan and normal activity including mobility and pharyngeal pumping in *C. elegans*, and it also improved the exercise capacity in mice with age-related decline of muscle function (*Ryu et al., 2016*). Similarly, rapamycin has been also found to increase the lifespan in worms (*Robida-Stubbs et al., 2012*), yeast (*Medvedik et al., 2007*), and flies (*Bjedov et al., 2010*), as well as mean and maximum lifespans in mice (*Harrison et al., 2009*). Metformin, a first-line drug for type 2 diabetes treatment, has been widely studied to extend the lifespan both in *C. elegans* (*Cabreiro et al., 2013*; *De Haes et al., 2014*; *Chen et al., 2017*) and mice (*Anisimov et al., 2008*; *Martin-Montalvo et al., 2013*).

In order to discover novel anti-aging compounds, we screened our in-house 1072 marketed drugs using *C. elegans* as an animal model for their ability to extend the lifespan. As marketed drugs generally have definite pharmacokinetics and pharmacodynamics properties and are useful for drug repurposing, our research group is focused on searching compounds for drug repurposing. Herein, in this study, the approved drug crotamiton has been found to show anti-aging activity for the first time and was further selected for the structural optimization.

Crotamiton is an inhibitor of TRPV4 (Transient Receptor Potential Vanilloid-4) channel and has been used as anti-scabies and anti-itch agent in humans for nearly 70 years (*Kittaka et al., 2017*). TRPV subfamily proteins are encoded by five genes in *C. elegans*, including *osm-9* (<u>osm</u>otic avoidance abnormal), *ocr-1* (<u>o</u>sm-9/<u>c</u>apsaicin receptor <u>r</u>elated), *ocr-2*, *ocr-3* and *ocr-4* (*Xiao and Xu, 2009*). Only loss of *osm-9* or *ocr-2* in worms resulted in the lifespan extension (*Sheng et al., 2017*). OSM-9 and OCR-2 can form heterotetrameric channels which transduce signals from olfactory, nociceptive, and serotonergic neurons (*Tobin et al., 2002*; *Ohnishi et al., 2020*; *Zhang et al., 2004*); however, the role of OSM-9 and OCR-2 in the regulation of stress resistance involves different mechanisms (*Moriuchi et al., 2018*). It has been shown in previous studies that the inactivation of OCR-2 extends the L1 starvation survival, while null mutations in *osm-9* did not alter L1 starvation survival (*Lee and Ashrafi, 2008*). *Osm-9* null mutants showed more resistance to oxidative or hypertonic stress than the control worms (*Lee et al., 2016*). Noticeably, it was reported that taurine, an essential amino acid involved in various physiological functions, promoted longevity of *C. elegans* in oxidative stress condition by inhibiting OSM-9 but not OCR-2 (*Moriuchi et al., 2018*). Taken together, further extensive research is still needed to decipher the downstream signaling pathways after OSM-9 or OCR-2 activation.

With an aim to get a potential anti-ageing tool molecule, structural optimization based on crotamiton led to the identification of JM03. This molecule displayed better activity than crotamiton in terms of the lifespan extension and stress resistance of *C. elegans*. To further decipher the mechanisms of JM03 involved in the anti-aging activity, this study was conducted with special emphasis on its interaction with OSM-9 or OCR-2.

## Results

### Crotamiton prolongs lifespan of *C. elegans*

To identify candidate anti-aging compounds, we initially performed a phenotypic screening of our in-house 1072 marketed drugs with 15 worms per concentration (100 μM) using an *C. elegans* model for their abilities of lifespan extension (*Figure 1—source data 1*). Thereafter, 125 drugs which showed up to 10% increase in mean lifespan extension as compared to the controls were selected for the secondary screening with 30 worms per concentration (100 μM) (*Figure 1—source data 2*). Finally, 10 drugs which showed up to 10% increase in mean lifespan extension were chosen for the third screening with 60 worms per concentration (400 μM, 100 μM, and 25 μM), respectively (*Figure 1—source data 2*). Apart from recently reported verapamil hydrochloride (*Liu et al., 2020*) and chlorpropamide (*Mao et al., 2022*) listed in our screening results, crotamiton was finally selected as one hit compound with significant effect on *C. elegans* lifespan extension in this study (p < 0.01) (*Figure 1a*). The distribution of approved drugs combinations per disease area in this phenotypic screening study was showed in *Figure 1b*. The safety of crotamiton was evaluated by following parameters: (1) The

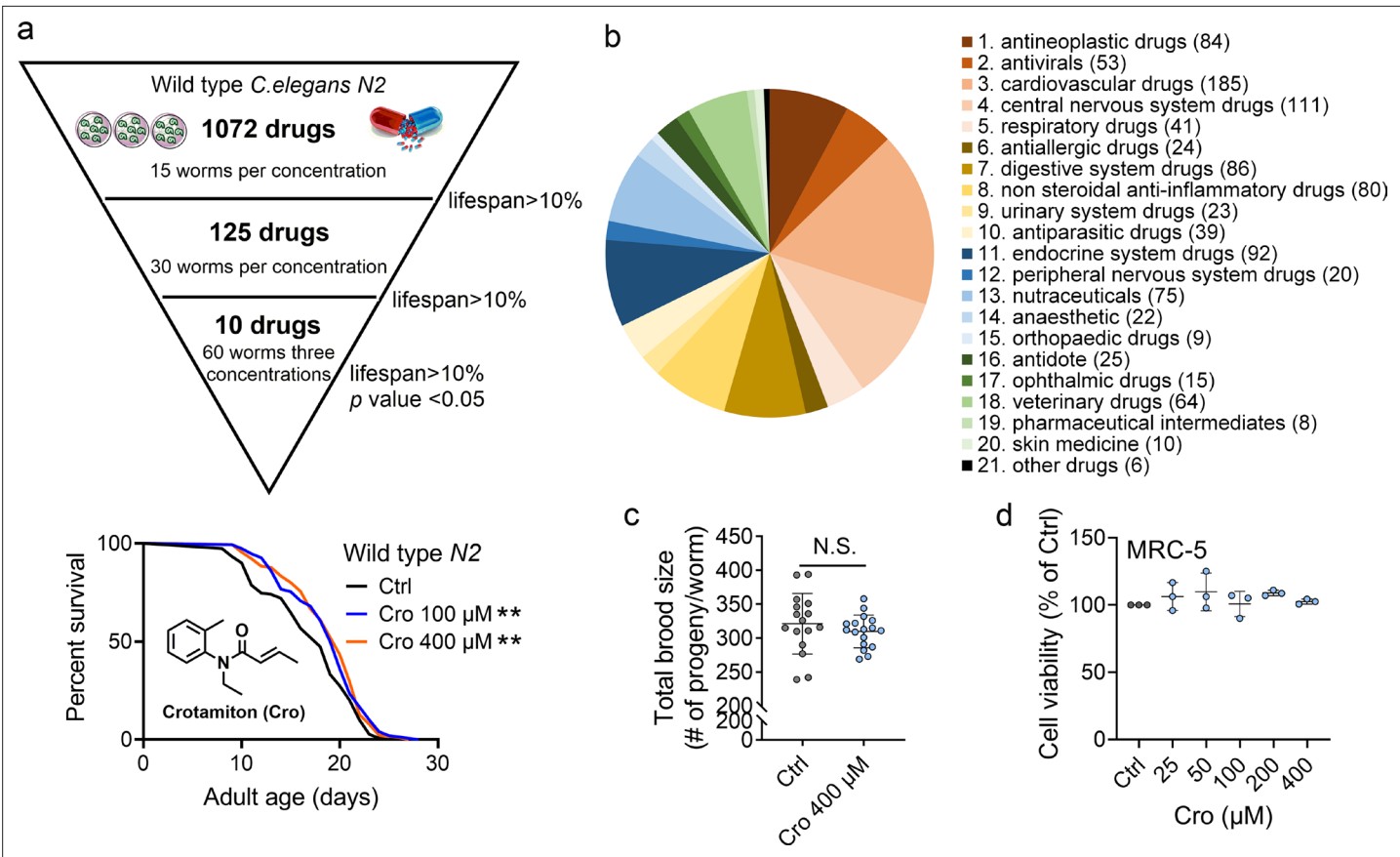

**Figure 1.** Crotamiton extends the lifespan of *Caenorhabditis elegans*. (**a**) Phenotypic screening led to the discovery of crotamiton as a hit compound for prolonging the lifespan in wild type (N2) worms. Data were compared using the Log-rank test. p = 0.0032 for Cro 100 μM. p = 0.0011 for Cro 400 μM. (**b**) The pie charts show the distribution of approved drug combinations per disease area in this phenotypic screening study. (**c**) The total brood size of crotamiton-treated *N2* worms. Control n = 16 and crotamiton n = 17. (**d**) The viability of crotamiton-treated MRC-5 cells. (**c**) Data have been represented as the mean ± SD, and comparisons are made using Student t-test. (**d**) Data have been represented as the mean ± SD, and comparisons are made using one-way ANOVA test. The graphics represent a compilation of at least three independent experiments. ** p < 0.01.

The online version of this article includes the following source data and figure supplement(s) for figure 1:

**Source data 1.** Lifespan data in the 1st round screening.

**Source data 2.** Lifespan data in the 2nd and 3rd round screening.

**Source data 3.** Lifespan data for (**a**), brood size for (**c**), and cell viability for (**d**).

**Figure supplement 1.** The anti-scabies drugs permethrin and benzyl benzoate failed to extend the lifespan in *Caenorhabditis elegans*.

**Figure supplement 1—source data 1.** Lifespan data of worms treated with permethrin or benzyl benzoatee.

**Figure supplement 2.** Crotamiton and JM03 did not reduce the bacterial growth at 400 μM concentration.

**Figure supplement 2—source data 1.** OD600 for OP50 culture measured every 12 hr in 72 hr.

reproductive capacity was not changed in crotamiton-treated worms at 400 μM (*Figure 1c*); (2) For the normal human fetal lung fibroblasts cells MRC-5, crotamiton showed no toxicity even up to 400 μM (*Figure 1d*).

In order to exclude the possibility that the anti-aging effect of crotamiton was related to its resistance to scabies, two anti-scabies drugs, permethrin, a sodium channel inhibitor and benzyl benzoate, which exerts toxic effects on the nervous system of scabies (*Sunderkötter et al., 2021*), were also examined along with crotamiton in the same experiment. However, both of the compounds failed to extend the worm lifespan (*Figure 1—figure supplement 1*), which indicated that the lifespan-extension of crotamiton was not related to its resistance to scabies, but closely related to its structure and target. Moreover, crotamiton showed no significant effect on the bacterial growth, which further

ascertained that the anti-aging effect is not the result of insufficient food (*Figure 1—figure supplement 2a*).

## JM03, the derivative of crotamiton, has better life extension activity in *C. elegans*

Structural optimization of crotamiton was conducted to identify more potent compounds with better anti-aging activity. The synthetic route for compounds JM01-JM15 was shown in *Figure 2a*. Treatment of the substituted *N*-ethylaniline with acryloyl chloride derivatives and potassium carbonate in dichloromethane at room temperature resulted in the formation of JM01-JM05, JM10, JM12, JM13, JM15, **9–11** in about 90–98% yield. Then, compounds JM10, JM12, JM15, **9–11** were conveniently hydrolyzed to provide the corresponding acid JM06-JM09, JM11, JM14.

Studies on the relationship between structure and activity (*Figure 2—figure supplement 1*) showed the removal of the methyl group from the *ortho*-position (crotamiton) to *meta*-position (JM01) led to the minor improvement in the activity for the R1 substituent group at benzene ring. However, moving the methyl group to the *para*-position (JM02) resulted in better activity. Replacing the *ortho*-methyl with *ortho*-fluoro (JM03), chloro (JM04), and bromo (JM05) significantly increased the activity. Additionally, incorporation of the carboxyl (JM06, JM07) on the benzene ring did not increase the activity. Since the introduction of a carboxyl at the terminal of alkenyl of crotamiton (JM08) improved the activity, we conducted additional modification based on the potent compounds JM02 and JM03. Unfortunately, no remarkable increased activity was observed after this step (*Figure 2—figure supplement 2*). Moreover, the movement of the fluoro substituent from the *ortho*-position (JM03) to the *para*-position (JM13) had no effect on the activity. Further adding carboxyl (JM14) or ethoxycarbonyl (JM15) was found to be detrimental. Considering the introduction of fluorine substituents into drugs can enhance biological activity and increase chemical or metabolic stability (*Hagmann, 2008*), JM03 was selected for the following study. The lifespan of worms treated with JM03 increased significantly as compared to those treated with crotamiton (p < 0.01, *Figure 2b*). Since 5-Fluorodeoxyuridine (FUdR) was used as contraceptive in our lifespan assay and is known to impact lifespan, we also tested the lifespan-extension effect of crotamiton and JM03 without FUdR and found two compounds could both prolong the lifespan of *C. elegans* in the absence of FUdR (*Figure 2c*). It has been reported that aging would lead to slower and uncoordinated body movement in *C. elegans* (*Herndon et al., 2002*). Therefore, keeping this in view, we measured the age-dependent muscle deterioration and diminished pharyngeal pumping rate in worms to assess the healthspan of worms treated with JM03. It was found that JM03 did not change the body bend rate of *C. elegans* at different age (*Figure 2d*). JM03-treated groups exhibited increased pharyngeal pumping rate at day 9 (*Figure 2e*). Additionally, no changed reproductive capacity was observed in JM03-treated worms (*Figure 2f*). Similar to crotamiton, the antiaging effect of JM03 is not the result of insufficient food (*Figure 1—figure supplement 2*).

## JM03-induced extension of lifespan depends on OSM-9 in *C. elegans*

Crotamiton is an inhibitor of TRPV4 channel (*Kittaka et al., 2017*), which shows similarity to the *C. elegans* channels OSM-9 (26% amino acid identity, 44% identity or conservative change) and OCR-2 (24% identity, 38% identity or conservative change) (*Liedtke et al., 2003*). In *C. elegans*, lacking of TRPV channel OSM-9 or OCR-2 resulted in the lifespan extension (*Riera et al., 2014*). Therefore, we further performed the lifespan analysis on *osm-9* or *ocr-2* knockdown worms to investigate the mechanism of JM03. As shown in *Figure 3a and b*, knockdown of *osm-9* or *ocr-2* via RNAi (*Figure 3d and e*) extended the lifespan of worms compared to the empty vector group. This indicated that TRPV inhibition extended *C. elegans* lifespan. Notably, JM03 failed to extend the lifespan of *osm-9* knockdown worms (*Figure 3a*), but still extended the lifespan of *ocr-2* knockdown worms (*Figure 3b*). Consistently, JM03 was found to be unresponsive to the lifespan of *osm-9(ky10)* mutants (*Figure 3c*). These results suggested that OSM-9 not OCR-2, played a leading role in JM03-mediated longevity.

## JM03 improves the ability of nematode to resist oxidative and hypertonic stress through OSM-9

It has been previously shown that the loss of OSM-9 enhanced the resistance of nematodes to the oxidative and hypertonic stress (*Lee et al., 2016*). Therefore, we also evaluated the efficacy of JM03

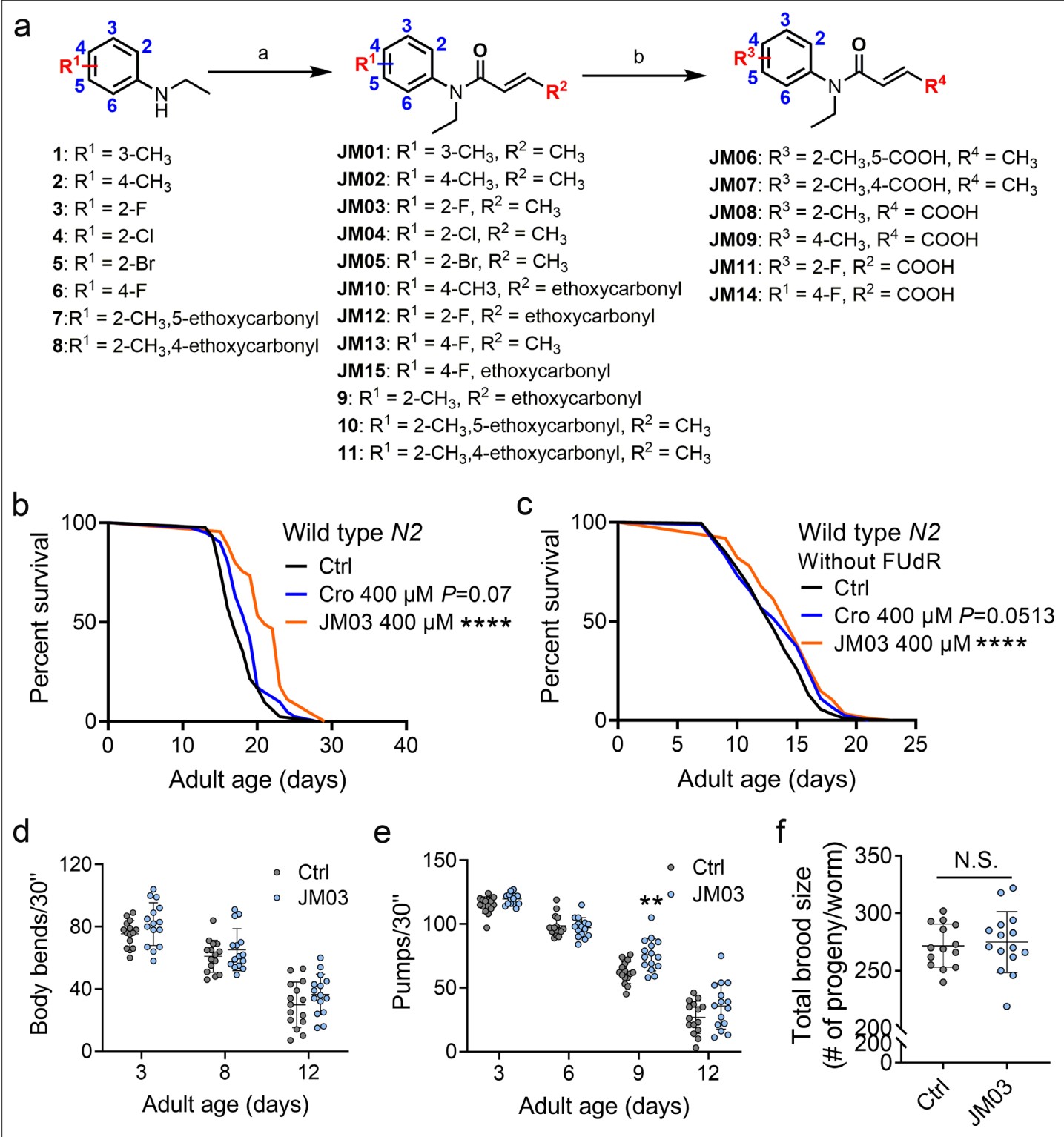

**Figure 2.** JM03 has better lifespan extension activity in *Caenorhabditis elegans*. (**a**) Synthesis of compounds JM01–JM15. Reagents and conditions: a. Acryloyl chloride derivatives, K$_2$CO$_3$, CH$_2$Cl$_2$, 0 °C to rt; b. 1 M NaOH (aq.), CH$_3$OH, rt. (**b**) JM03 prolonged lifespan in wild-type worms. p-values by Log-rank test. p = 0.0737 for Cro 400 μM. p < 0.0001 for JM03 400 μM. (**c**) JM03 prolonged lifespan in wild-type worms without FUdR treatment. p-values by Log-rank test. p = 0.0513 for Cro 400 μM. p = 0.0003 for JM03 400 μM. (**d**) The mobility of JM03-treated *N2* worms by analyzing the body bend rate at days 3, 8, and 12. Control n = 15 and JM03 n = 15. (**e**) The pharyngeal pumping rate of JM03-treated *N2* worms. Control n = 15 and JM03 n = 15 at days 3, 6, 9, and 12. p-values by two-way AVOVA. p = 0.0015 for 9 days. (**f**) The total brood size of JM03-treated *N2* worms. Control n = 14 and JM03 n =

*Figure 2 continued on next page*

*Figure 2 continued*

15. (**b–c**) Data are compared using the Log-rank test. (**d-e**) Data have been represented as the mean ± SD, and comparisons are made using two-way AVOVA. (**f**) Data have been represented as the mean ± SD, and comparisons are made using Student t-test. The graphics represent a compilation of at least three independent experiments. ** p < 0.01, **** p < 0.0001.

The online version of this article includes the following source data and figure supplement(s) for figure 2:

**Source data 1.** Lifespan data for (b); number of body bends for (c); number of pumps in 30″ for (d); brood size for (e) and cell viability for (f).

**Figure supplement 1.** The structures and mean percentage of lifespan extension by crotamiton derivatives.

**Figure supplement 1—source data 1.** Lifespan data of crotamiton derivatives.

**Figure supplement 2.** The structures and mean percentage of lifespan extension by crotamiton derivatives.

**Figure supplement 2—source data 1.** Lifespan data of crotamiton derivatives.

under the oxidative or hypertonic stress condition. As shown in *Figure 4a*, the lifespan of *C. elegans* under paraquat-induced oxidative stress condition was significantly increased in JM03-treated group compared with control or crotamiton-treated group. Then, we examined whether the effect of JM03 under oxidative stress condition is mediated via OSM-9 and OCR-2. It was shown that JM03 treatment did not increase the lifespan *osm-9(ky10)* mutants (*Figure 4b*), but increased the lifespan of

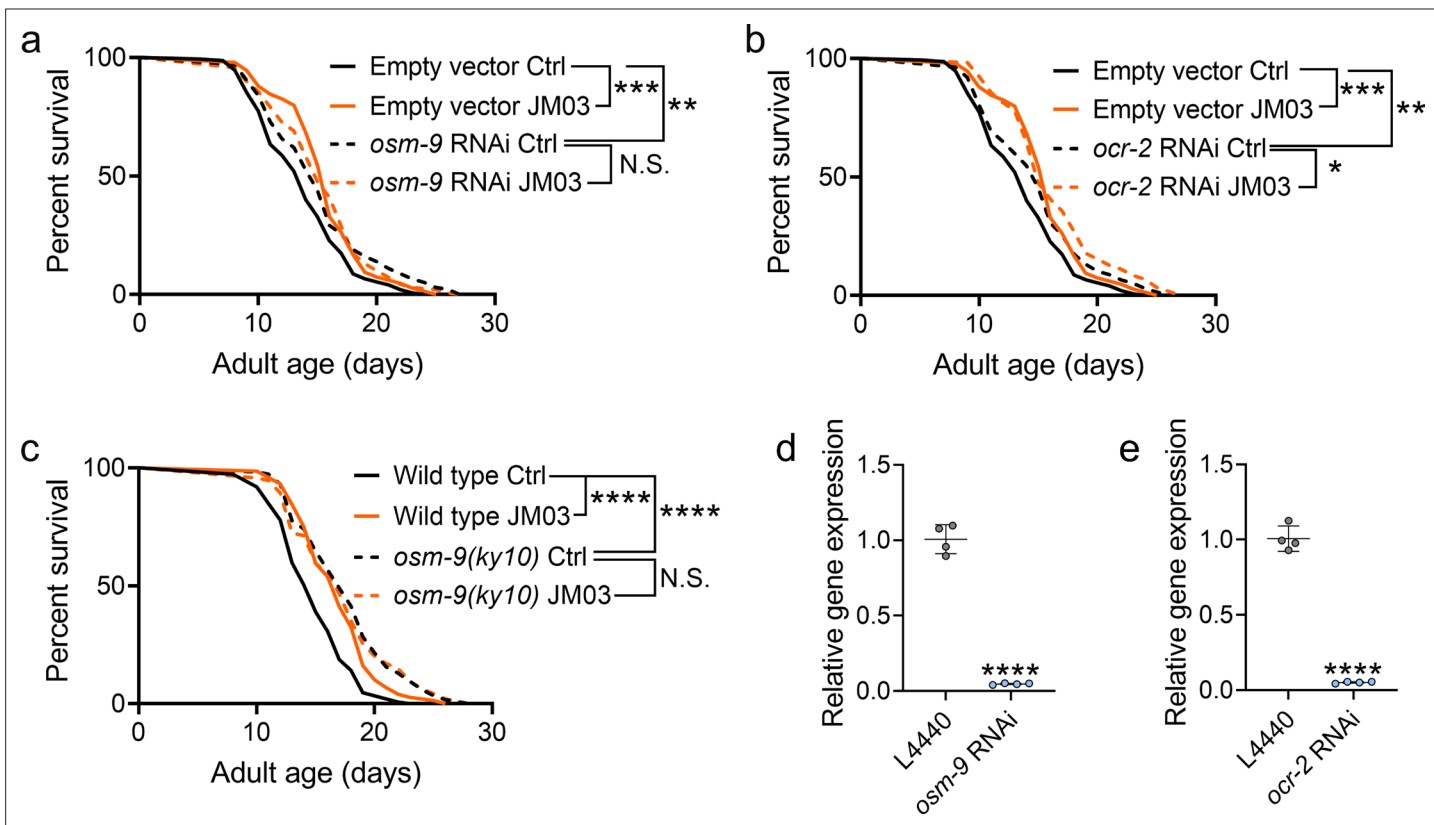

**Figure 3.** JM03-induced lifespan extension depends on OSM-9. (**a**) JM03 failed to extend the lifespan of *osm-9* RNAi worms. p-values by Log-rank test. p = 0.0002 between Empty vector Ctrl and Empty vector JM03. p = 0.0028 between Empty vector Ctrl and *osm-9* RNAi Ctrl. (**b**) JM03 extended the lifespan of *ocr-2* RNAi worms. p-values by Log-rank test. p = 0.0002 between Empty vector Ctrl and Empty vector JM03. p = 0.0084 between Empty vector Ctrl and *ocr-2* RNAi Ctrl. p = 0.0259 between *ocr-2* RNAi Ctrl and *ocr-2* RNAi JM03. (**c**) JM03 failed to extend the lifespan of *osm-9(ky10)* mutants. p-values by Log-rank test. p < 0.0001 for wild-type JM03 and *osm-9(ky10)* Ctrl. (**d**) The transcriptional level of *osm-9* decreased after RNAi treatment. p-values by Student t-test. p < 0.0001 for *osm-9* RNAi. (**e**) The transcriptional level of *ocr-2* decreased after RNAi treatment. p-values by Student t-test. p < 0.0001 for *ocr-2* RNAi. (**a–c**) Data are compared using the Log-rank test. (**d-e**) Data have been represented as the mean ± SD, and comparisons are made using Student t-test. The graphics represent a compilation of at least three independent experiments. * p < 0.05, ** p < 0.01, *** p < 0.001, **** p < 0.0001. N.S., not significant.

The online version of this article includes the following source data for figure 3:

**Source data 1.** Lifespan data for (**a–c**) and relative gene expression for (**d, e**).

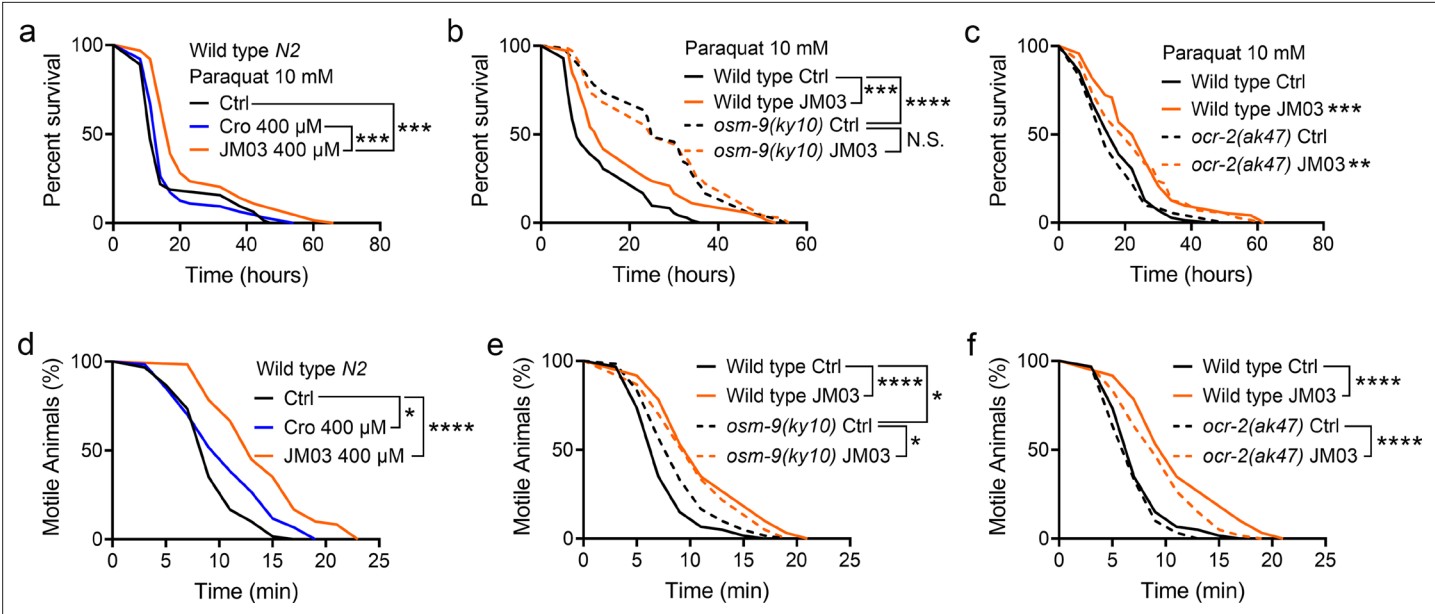

**Figure 4.** OSM-9 inhibition induced by JM03 has beneficial effect for *Caenorhabditis elegans* lifespan under oxidative and hypertonic stress conditions. (**a**) JM03 extended the lifespan of wild-type (**N2**) worms under paraquat-induced oxidative stress condition. p-values by Log-rank test. p = 0.0001 between Ctrl and JM03 400 µM. p = 0.0002 between Cro 400 µM and JM03 400 µM. (**b**) JM03 failed to extend the lifespan of *osm-9(ky10)* mutants under oxidative stress condition. p-values by Log-rank test. p = 0.0009 between Wild-type Ctrl and Wild-type JM03. p < 0.0001 between Wild-type Ctrl and *osm-9(ky10)* Ctrl. (**c**) JM03 extended the lifespan of *ocr-2(ak47)* mutants under oxidative stress condition. p-values by Log-rank test. p = 0.0005 between Wild-type Ctrl and Wild-type JM03. p = 0.0024 between *ocr-2(ak47)* Ctrl and *ocr-2(ak47)* JM03. (**d**) JM03 significantly reduced the paralysis for wild-type worms under NaCl-induced hypertonic stress condition. p-values by Log-rank test. p = 0.0171 between Ctrl and Cro 400 µM. p < 0.0001 between Ctrl and JM03 400 µM. (**e**) JM03 reduced the responsiveness for *osm-9(ky10)* mutants under hypertonic stress condition. p-values by Log-rank test. p < 0.0001 between Wild-type Ctrl and Wild-type JM03. p = 0.0128 between Wild-type Ctrl and *osm-9(ky10)* Ctrl. p = 0.0254 between *osm-9(ky10)* Ctrl and *osm-9(ky10)* JM03. (**f**) JM03 significantly reduced the paralysis for *ocr-2(ak47)* mutants similar to wild-type worms under hypertonic stress condition. p-values by Log-rank test. p < 0.0001 between Wild-type Ctrl and Wild-type JM03. p < 0.0001 between *ocr-2(ak47)* Ctrl and *ocr-2(ak47)* JM03. (**a–f**) Data are compared using the Log-rank test. The graphics represent a compilation of at least three independent experiments. * p < 0.05, ** p < 0.01, *** p < 0.001, **** p < 0.0001. N.S., not significant.

The online version of this article includes the following source data for figure 4:

**Source data 1.** lifespan data for (**a–c**) and relative gene expression data for (**d, e**).

ocr-2(ak47) mutants under paraquat-induced oxidative stress condition (***Figure 4c***), which suggested OSM-9 is required for JM03 to improve the anti-oxidative stress ability.

In addition, the motile time of *C. elegans* under hypertonic stress condition revealed by motility assays was significantly increased in JM03-treated group compared with control or crotamiton-treated group (***Figure 4d***). We also examined whether the effect of JM03 under hypertonic stress condition is mediated via OSM-9 and OCR-2. *Osm-9(ky10)* mutants exhibited increased motility and viability upon prolonged exposures to high osmotic environments compared with wild-type *N2* (***Figure 4e***), while *ocr-2(ak47)* mutants exhibited motility and viability similar to wild-type *N2* (***Figure 4f***). For o*sm-9(ky10)* mutants, the increased significance of the motile time under hypertonic stress condition is reduced after JM03 treatment (***Figure 4e***). But for *ocr-2(ak47)* mutants, JM03 still significantly increased the motile time of *C. elegans* under hypertonic stress condition (***Figure 4f***). Taken together, these results suggested that the OSM-9 inhibition by JM03 increased the anti-oxidative and anti-hypertonic stress ability of *C. elegans*.

## JM03 upregulated the genes associated with proteostasis

Similar to crotamiton, JM03 also showed no toxicity against MRC-5 cells even up to 400 µM (***Figure 5a***). Then we investigated the effect of JM03 on cellular proteostatic modules. We observed that 72 hr cell exposure to JM03 induced a mild upregulation of molecular chaperones genes (*Clusterin*), and a significant upregulation of Nrf2 transcriptional targets genes (*Keap1, Nqo1, Txnrd1*) (***Figure 5b***). It has been reported that oxidative (***Mark et al., 2016***) and hypertonic (***Lee et al., 2016***) stress enhance

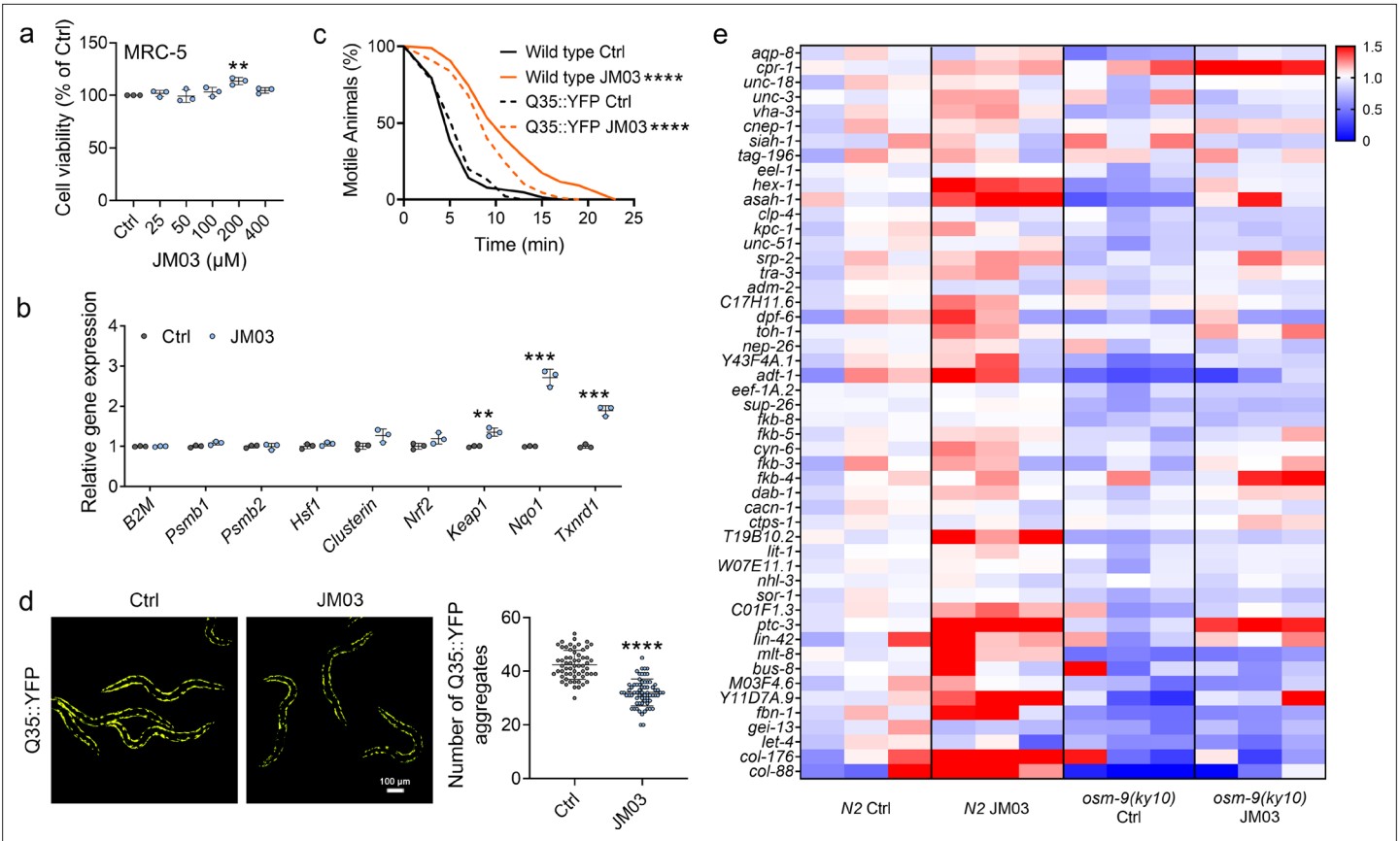

**Figure 5.** OSM-9 inhibition induced by JM03 has beneficial effect for *Caenorhabditis elegans* lifespan under hypertonic stress conditions. (**a**) The viability of JM03-treated MRC-5 cells. p-values by one-way ANOVA test. p = 0.0036 for JM03 200 μM. (**b**) JM03 significantly increased the transcriptional expression of proteostasis related genes in MRC-5 cells. p-values by Student t-test. p = 0.0041 for *Keap1*. p = 0.0001 for *Nqo1*. p = 0.0003 for *Txnrd1*. (**c**) JM03 significantly reduced the paralysis for Q35::YFP worms similar to wild-type worms under hypertonic stress condition. p-values by Log-rank test. p < 0.0001 between Wild-type Ctrl and Wild-type JM03. p < 0.0001 between Q35::YFP Ctrl and Q35::YFP JM03. (**d**) JM03 significantly reduced the Q35::YFP aggregation. p-values by Student t-test. p < 0.0001 for JM03 treatment. (**e**) Putative proteostasis genes differentially upregulated by JM03 treatment in *N2* control worms but not *osm-9(ky10)* worms by transcriptome analysis. (**a**) Data have been represented as the mean ± SD, and comparisons are made using one-way ANOVA test. (**b, d**) Data have been represented as the mean ± SD, and comparisons are made using Student t-test. (**c**) Data are compared using the Log-rank test. The graphics represent a compilation of at least three independent experiments. ** p < 0.01, *** p < 0.001, **** p < 0.0001.

The online version of this article includes the following source data and figure supplement(s) for figure 5:

**Source data 1.** Cell viability for (**a**); relative gene expression data for (**b**); lifespan data for (**c**); number of Q35::YFP aggregates of each worm for (**d**); RNA sequencing data of proteostasis-related genes for (**e**).

**Figure supplement 1.** The principal component analysis (PCA) of the transcriptome analysis by RNA-sequencing.

rapid and widespread protein aggregation and misfolding in *C. elegans*. Here, we investigated the efficacy of JM03 to reduce the aggregation of protein using Q35::YFP, a worm strain expressing poly-glutamine (Q35) containing yellow fluorescent (YFP) protein in their body wall muscle. Q35::YFP is normally fully soluble in the muscles cells of young worms, but undergoes a slow, progressive aggregation as *C. elegans* ages (*Morley et al., 2002*). Here, we observed the anti-hypertonic stress ability of Q35::YFP and wild-type worms was similar (*Figure 5c*). JM03 significantly increased the motile time of Q35::YFP or wild-type worms exposed to 500 mM NaCl, which is consistent with results shown in *Figure 4d*. Meanwhile, Q35::YFP aggregation was significantly reduced when treated with JM03 (*Figure 5d*), which suggested JM03 reduced the aggregation of protein in *C. elegans*.

To gain a more detailed picture of the genetic expression after JM03 treatment, worms fed with JM03 or DMSO for 10 days were processed for RNA-sequencing (RNA-seq) to analyze the altered mRNA abundance (GEO accession number: GSE200459). In order to test the sequencing quality of RNA-seq data, principal component analysis (PCA) was performed based on the differentially

expressed genes between different types of samples. As shown in *Figure 5—figure supplement 1*, four clusters were clearly defined and the RNA-seq data of the same samples tend to congregate in groups, which indicated the good RNA-seq quality. It was reported that the improved proteostasis capacity of the *osm-9* null mutant was due to altered expression of genes encoding components of the proteostasis network (including protein degradation, protein synthesis, protein folding and so on) (*Lee et al., 2016*). Interestingly, JM03 can upregulate the expression of the genes that play known or presumptive roles in proteostasis in *N2* worms but not in *osm-9(ky10)* mutant revealed by RNA-seq results (*Figure 5e*). Taken together, these results supported the notion that JM03 upregulates the genes associated with proteostasis through OSM-9 leading to enhanced proteostasis capacity, which may improve the ability of nematode to resist oxidative stress and hypertonic stress.

## JM03 activates the SKN-1 stress response pathway in *C. elegans*

The transcription factors DAF-16 (*Baxi et al., 2017*; *Li et al., 2019a*) and SKN-1 play important roles in regulating stress resistance, longevity and proteostasis (*Blackwell et al., 2015*; *Jones et al., 2020*). Therefore, we examined the effect of JM03 on DAF-16 and SKN-1 pathway. JM03 prolonged the lifespan of *daf-16(mu86)* null mutant (*Figure 6—figure supplement 1*), suggesting that DAF-16 is not required for JM03-induced lifespan extension. On the contrary, JM03 did not prolonged the lifespan of *skn-1(zu135)* mutants with loss of function mutation in all SKN-1 isoforms (*Bishop and Guarente, 2007*), indicating that SKN-1 played an essential role in JM03-induced positive effects (*Figure 6a*). Given the dependency of the transcription factor SKN-1 in JM03-induced lifespan extension, we further examined our RNA-seq dataset to determine whether expression of target genes of SKN-1 might be perturbed by JM03 treatment. We found that *skn-1* and its target genes, such as *gst-4*, *gst-6*, *gst-7*, *gcs-1*, *prdx-3*, and *mtl-1* were upregulated by JM03 (*Figure 6b*).

Next, we examined the effect of JM03 on the activation of the SKN-1 stress response pathway using a previously described GFP translational reporter fused to the *skn-1* promoter (*Bishop and Guarente, 2007*; *Kahn et al., 2008*). JM03 treatment significantly increased the intensity of GFP fluorescence driven by the native *skn-1* promoter (*ls007[skn-1::gfp]*) (*Bishop and Guarente, 2007*; *Figure 6c*). Concurrently, it also significantly increased the transcriptional expression of *skn-1* itself and *skn-1* regulated genes *gst-4*, *gst-6*, *gst-7*, *gcs-1*, *ctl-2*, *prdx-3* or *mtl-1* (*Figure 6d*). Subsequently, we also confirmed the increased expression of glutathione S-transferase-4 (*gst-4*), a key downstream target of SKN-1 (*Li et al., 2019b*), based on the GFP fluorescence signal of *gst-4::gfp* worms (*Figure 6e*). In addition, JM03 did not extend the lifespan of *skn-1(zu135)* mutants under oxidative stress condition (*Figure 6f*). In conclusion, JM03 prolongs the lifespan and improves stress-resistance ability of *C. elegans* through SKN-1 pathway.

## Discussion

Drug repurposing has emerged as an effective approach for the rapid identification and development of pharmaceutical molecules with novel activities against various diseases based on the already known marketed drugs (*Pushpakom et al., 2019*). Herein, we explored the possibility of identifying the potent drugs which could increase the longevity by screening our in-house marketed drugs using lifespan extension assays in *C. elegans*. In the first round, a large-scale anti-aging drug screening was conducted with the sample size of 15 animals per drug at a single concentration and only the cohorts with +10% or over increase in lifespan made them the second round. The power of detection of the first-round screen is roughly less than 15% based on the power calculation tables provided by recent computational analysis of lifespan experiment reproducibility (*Petrascheck and Miller, 2017*). In addition, all drugs were screened in the first and second round at a single concentration (100 μM) and this concentration is not modified based on the compound of interest. The concentration optimization was only performed in the third-round screen for the confirmed 10 hit compounds. Therefore, our screening strategy indeed brought a number of false negatives and miss a substantial number of anti-aging compounds.

After three rounds of screening, crotamiton, which was known as an anti-scabies and anti-itch agent was found to increase the lifespan. It has been reported that crotamiton is an inhibitor of human TRPV4 channel (*Kittaka et al., 2017*), which is homologous to OSM-9 and OCR-2 channels in *C. elegans* (*Xiao and Xu, 2009*). Loss of OCR-2 or OSM-9, can result in the lifespan extension in *C.*

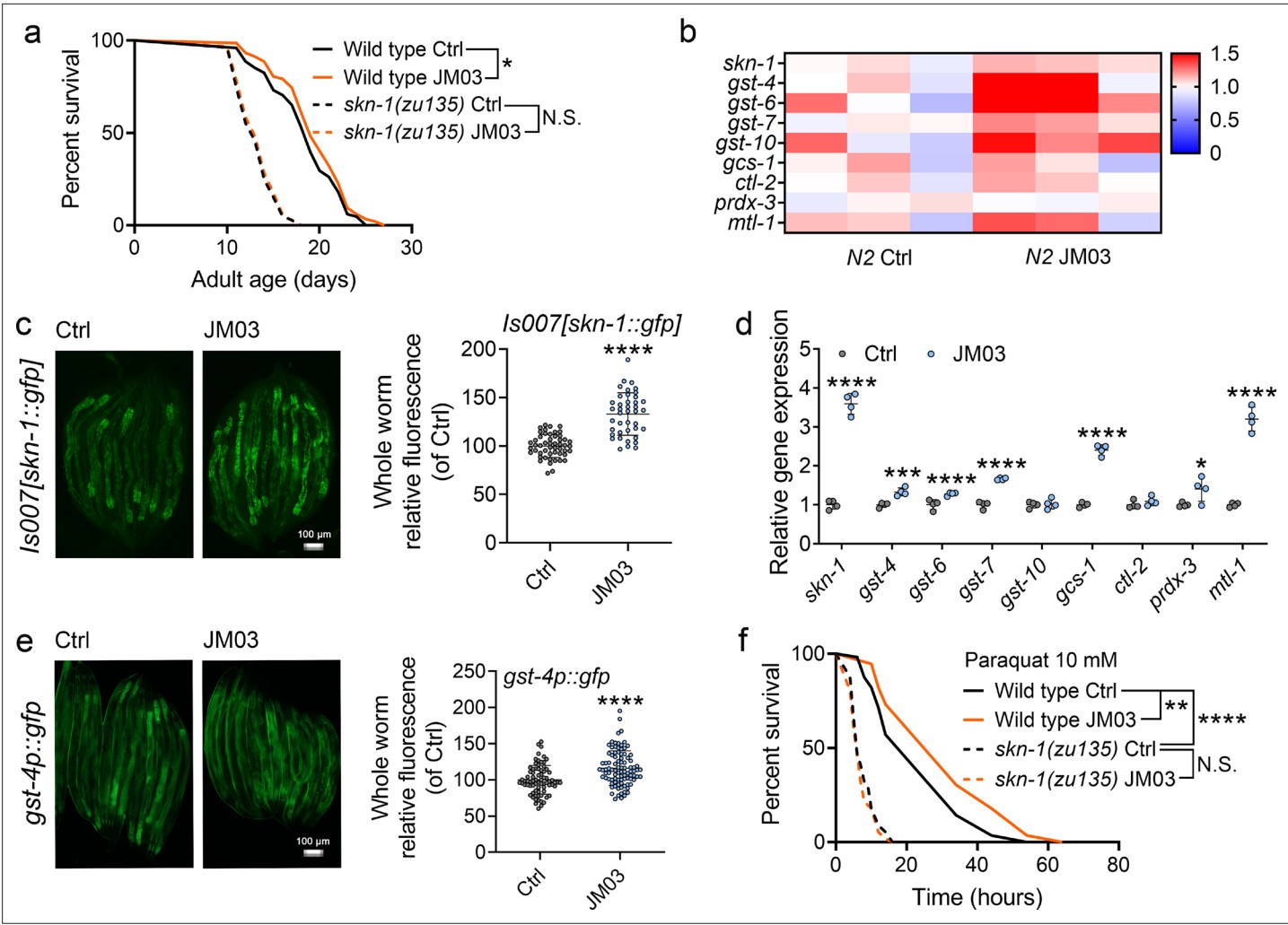

**Figure 6.** JM03-induced lifespan extension through SKN-1 pathway. (**a**) JM03 treatment failed to extend the lifespan of *skn-1(zu135)* mutants. p-values by Log-rank test. p = 0.0569 between Wild-type Ctrl and Wild-type JM03. (**b**) *Skn-1* and its targets genes upregulated by JM03 in wild-type worms by transcriptome analysis. (**c**) JM03 significantly increased the fluorescence intensity of *skn-1::gfp*. Scale bar = 100 μm. p-values by Student t-test. p < 0.0001 for JM03 treatment. (**d**) JM03 significantly increased the transcriptional expression of *skn-1* and *skn-1* regulated genes. p-values by Student t-test. p < 0.0001 for *skn-1*. p = 0.0006 for *gst-4*. p < 0.0001 for *gst-6*. p < 0.0001 for *gst-7*. p < 0.0001 for *gcs-1*. p = 0.0424 for *prdx-3*. p < 0.0001 for *mtl-1*. (**e**) JM03 significantly upregulated the fluorescence intensity of *gst-4p::gfp*. Scale bar = 100 μm. p-values by Student t-test. p < 0.0001 for JM03 treatment. (**f**) JM03 treatment failed to extend the lifespan of *osm-9(ky10)* and *skn-1(zu135)* mutants under oxidative stress condition. p-values by Log-rank test. p = 0.0059 between Wild-type Ctrl and Wild-type JM03. p < 0.0001 between Wild-type Ctrl and *skn-1(zu135)* Ctrl. (**a, f**) Data are compared using the Log-rank test. The graphics represent a compilation of at least three independent experiments. (**c–e**) Data have been represented as the mean ± SD, and comparisons are made using Student t-test. * p < 0.05, ** p < 0.01, *** p < 0.001, **** p < 0.0001. N.S., not significant.

The online version of this article includes the following source data and figure supplement(s) for figure 6:

**Source data 1.** Lifespan data for (**a**); RNA-sequencing data of *Skn-1* and its targets genes for (**b**); mean fluorescence intensity of each worm for (**c**); relative gene expression data for (**d**); mean fluorescence intensity of each worm for (**e**); lifespan of each worm in paraquat for (**f**).

**Figure supplement 1.** JM03 treatment extended the lifespan of *daf-16(mu86)* mutants.

**Figure supplement 1—source data 1.** Lifespan data for wild type and *daf16* (mu86).

*elegans* (**Riera et al., 2014**). In our study, JM03 further increased the lifespan for the *ocr-2* knockdown *C. elegans*, but was ineffective for the knockdown or knockout of *osm-9* (**Figure 3**), which suggested JM03 selectively acted on OSM-9, not OCR-2. Furthermore, JM03 improved the antioxidant and anti-hyperosmotic stress resistance of wild type worms (**Figure 4a and d**). Interestingly, *osm-9* mutants showed enhanced ability to resist oxidative stress and hypertonic stress (**Figure 4b and e**), while *ocr-2* mutants did not (**Figure 4c and f**). These results also supported that JM03 selectively acted on

OSM-9, not OCR-2. Consistently, JM03 still had significant anti-oxidant and anti-hyperosmotic effects on *ocr-2* mutants (**Figure 4c and f**), but not *osm-9* mutants (**Figure 4b and e**). It is noted that OSM-9 is not the only mechanism that mediate the anti-hyperosmotic effect of JM03 because JM03 retained a slight effect on osmotic pressure resistance of the *osm-9* mutants.

OSM-9 plays major roles in transduction and regulation of signals in several sensory neurons and is important for processes such as volatile chemotaxis and osmotic avoidance (**Colbert et al., 1997**). *Osm-9* null mutant was reported to show enhanced survival in hypertonic environments, not due to altered systemic volume regulation or glycerol accumulation and instead may be due to enhanced proteostasis capacity (**Lee et al., 2016**). Consistently, JM03 treatment also enhanced proteostasis capacity in *C. elegans* revealed by reduced aggregation of Q35 (**Figure 5d**). Besides, JM03 can upregulate the expression of the genes that play known or presumptive roles in proteostasis in *N2* worms but not in *osm-9(ky10)* mutant by transcriptome analysis (**Figure 5e**). Among these genes, the increased expression of *aquaporin-8* (*aqp-8*) in *osm-9(ok1677)* mutant was also reported by a Germany lab (**Igual Gil et al., 2017**). Considering the essential roles of AQP-8 in sustaining the salt/ water balance in various cells types and tissues, the loss/inhibition of *osm-9* might help to maintain the salt/water balance to promote proteostasis during the response to hyperosmotic stress.

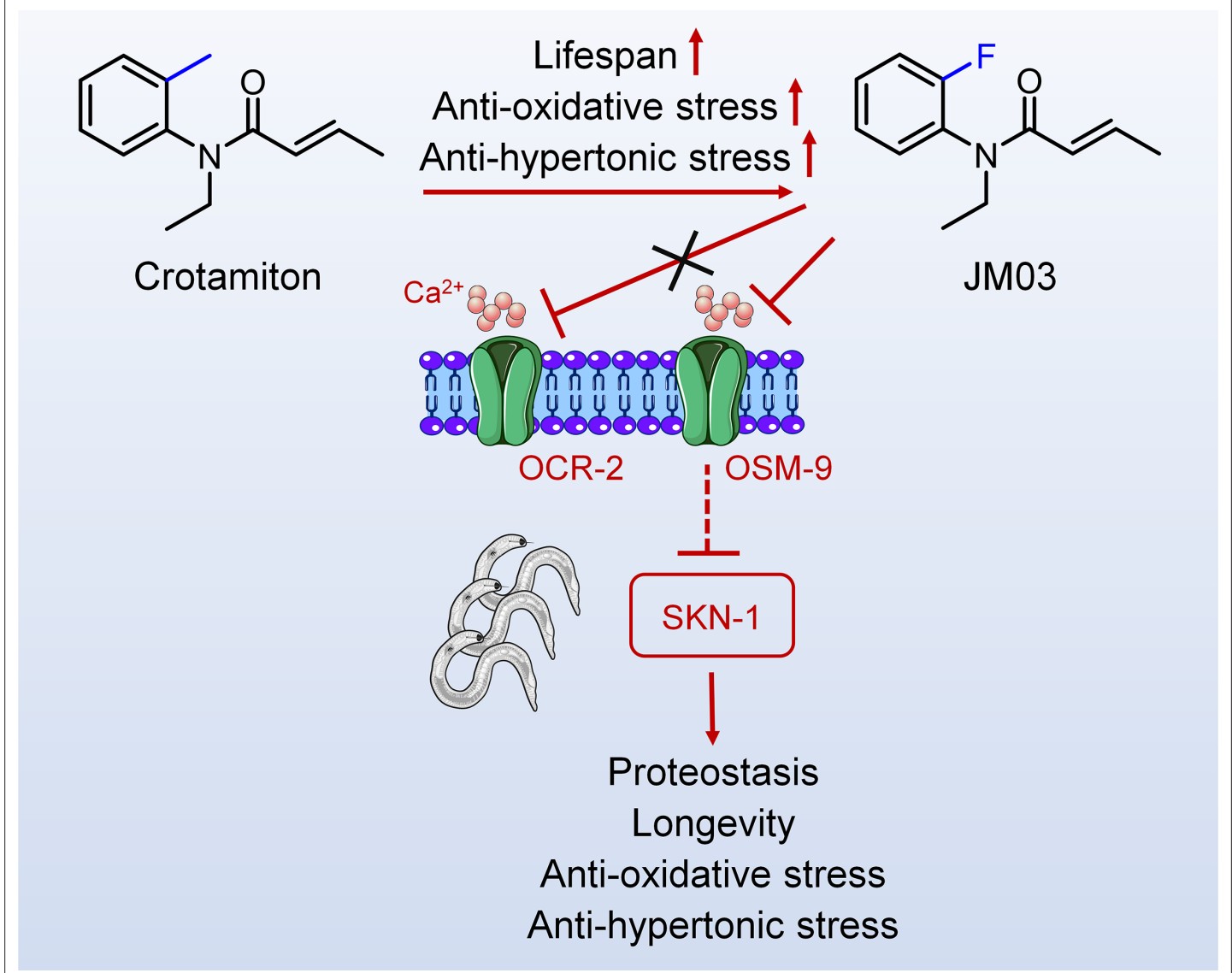

**Figure 7.** Schematic diagram of the mechanism of action of JM03 for regulating the lifespan, anti-oxidative and anti-hypertonic stress ability in *Caenorhabditis elegans*.

To investigate which downstream signaling was activated after OSM-9 inhibition by JM03, two important stress response transcription factors DAF-16 and SKN-1 were examined (**Mark et al., 2016**; **Alavez et al., 2011**; **Cohen et al., 2006**). Lifespan analysis showed that JM03 extended the lifespan through SKN-1 (**Figure 6a**), but not DAF-16 (**Figure 6—figure supplement 1**). Then our RNA-sequencing and qPCR data both showed the transcriptional expression of *skn-1* itself and *skn-1* regulated genes were significantly increased in JM03-treated worms (**Figure 6b and d**). The expression of SKN-1 and GST-4 was confirmed by using GFP translational reporter worms (*skn-1::gfp* and *gst-4::gfp*) (**Figure 6c and e**). These results provide the evidence that JM03 activates SKN-1 signaling that regulates longevity, stress resistance, and proteostasis. But how JM03 activates SKN-1 signaling after inhibiting OSM-9 remained to be studied.

In conclusion, overall results showed that JM03 increased the lifespan of *C. elegans* by inhibiting OSM-9, and then activated SKN-1, which improved proteostasis, stress resistance and lifespan extension in *C. elegans* (**Figure 7**). Since OSM-9 is the homologous to mammal TRPV channels, it's very interesting to examine whether JM03 acts selectively on a certain TRPV subtypes in mice in future studies.

# Materials and methods

## Key resources table

| Reagent type (species) or resource | Designation | Source or reference | Identifiers | Additional information |
|---|---|---|---|---|
| Commercial assay or kit | GMyc-PCR Mycoplasma Test Kit | Yeasen | Cat# 40,601ES10 | |
| Commercial assay or kit | cell counting kit-8 solution | Targetmol | Cat# C0005 | |
| Commercial assay or kit | Trizol Reagent | Beyotime | Cat# R0016 | |
| Commercial assay or kit | Total RNA Kit II | Omega | Cat# R6934-01 | |
| Commercial assay or kit | Hifair II 1st Strand cDNA Synthesis SuperMix | Yeasen | Cat# 11,120ES60 | |
| Commercial assay or kit | Hieff qPCR SYBR Green Master Mix | Yeasen | Cat# 11,201ES08 | |
| Cell line (*Homo-sapiens*) | MRC-5 cells | National Collection of Authenticated Cell Cultures | Cat# SCSP-5040 | |
| Strain, strain background (*Escherichia coli*) | OP50 | *Caenorhabditis* Genetics Center | | |
| Strain, strain background (*Escherichia coli*) | HT115 | *Caenorhabditis* Genetics Center | | |
| Strain, strain background (*Caenorhabditis elegans*) | N2 | *Caenorhabditis* Genetics Center | WormBase ID: WBStrain00000001 | wild type |
| Strain, strain background (*Caenorhabditis elegans*) | CX4544 | *Caenorhabditis* Genetics Center | WormBase ID: WBStrain00005264 | Genotype: *ocr-2(ak47) IV* |
| Strain, strain background (*Caenorhabditis elegans*) | CX10 | *Caenorhabditis* Genetics Center | WormBase ID: WBStrain00005214 | Genotype: *osm-9(ky10) IV* |
| Strain, strain background (*Caenorhabditis elegans*) | AM140 | *Caenorhabditis* Genetics Center | WormBase ID: WBStrain00000182 | Genotype: rmIs132 [unc-54p::Q35::YFP] |
| Strain, strain background (*Caenorhabditis elegans*) | LG333 | *Caenorhabditis* Genetics Center | WormBase ID: WBStrain00024182 | Genotype: *skn-1(zu135) (IV)/nT1[qIs51] (IV;V)*; ldIs7 [skn-1b/c::gfp] |
| Strain, strain background (*Caenorhabditis elegans*) | CF1038 | *Caenorhabditis* Genetics Center | WormBase ID: WBStrain00004840 | Genotype: *daf-16(mu86) (I)* |
| Strain, strain background (*Caenorhabditis elegans*) | CL2166 | *Caenorhabditis* Genetics Center | WormBase ID: WBStrain00005102 | Genotype: dvIs19 [gst-4p::gfp::NLS] (III) |
| Strain, strain background (*Caenorhabditis elegans*) | EU31 | *Caenorhabditis* Genetics Center | WormBase ID: WBStrain00007251 | Genotype: *skn-1(zu135) (IV)/ nT1[unc-?(n754);let-?] (IV;V)* |
| Sequence-based reagent | *B2M*-F | This paper | PCR primers | ACTGAATTCACCCCCACTGA |

*Continued on next page*

*Continued*

| Reagent type (species) or resource | Designation | Source or reference | Identifiers | Additional information |
|---|---|---|---|---|
| Sequence-based reagent | *B2M*-R | This paper | PCR primers | AAGCAAGCAAGCAGAATTTGG |
| Sequence-based reagent | *Psmb1*-F | This paper | PCR primers | GGATGCAGCGGTTTTCATGG |
| Sequence-based reagent | *Psmb1*-R | This paper | PCR primers | AATTGCCCCCGTAGTCATGG |
| Sequence-based reagent | *Psmb2*-F | This paper | PCR primers | CTGCTCCGCCCTCCATTAAC |
| Sequence-based reagent | *Psmb2*-R | This paper | PCR primers | GCCAAGCATGGAGTAGAACG |
| Sequence-based reagent | *Hsf-1*-F | This paper | PCR primers | TATGGCTTCCGGAAAGTGGT |
| Sequence-based reagent | *Hsf-1*-R | This paper | PCR primers | GGAACTCCGTGTCGTCTCTC |
| Sequence-based reagent | *Clusterin*-F | This paper | PCR primers | AAACGAAGAGCGCAAGACAC |
| Sequence-based reagent | *Clusterin*-R | This paper | PCR primers | TGTTTCAGGCAGGGCTTACA |
| Sequence-based reagent | *Nrf2*-F | This paper | PCR primers | CATCCAGTCAGAAACCAGTGG |
| Sequence-based reagent | *Nrf2*-R | This paper | PCR primers | GCAGTCATCAAAGTACAAAGCAT |
| Sequence-based reagent | *Keap1*-F | This paper | PCR primers | AGTTCATGGCCCACAAGGTG |
| Sequence-based reagent | *Keap1*-R | This paper | PCR primers | AATGGACACCACCTCCATGC |
| Sequence-based reagent | *Nqo1*-F | This paper | PCR primers | AGCAGACGCCCGAATTCAAA |
| Sequence-based reagent | *Nqo1*-R | This paper | PCR primers | AGAGGCTGCTTGGAGCAAAA |
| Sequence-based reagent | *Txnrd1*-F | This paper | PCR primers | TTGGAGTGCGCTGGATTTCT |
| Sequence-based reagent | *Txnrd1*-R | This paper | PCR primers | TTTGTTGGCCATGTCCTGGT |
| Sequence-based reagent | *ama-1*-F | This paper | PCR primers | TGGAACTCTGGAGTCACACC |
| Sequence-based reagent | *ama-1*-R | This paper | PCR primers | CATCCTCCTTCATTGAACGG |
| Sequence-based reagent | *act-1*-F | This paper | PCR primers | ATGTGTGACGACGAGGTTGC |
| Sequence-based reagent | *act-1*-R | This paper | PCR primers | ACTTGCGGTGAACGATGGATG |
| Sequence-based reagent | *skn-1*-F | This paper | PCR primers | ACAGTGCTTCTCTTCGGTAGC |
| Sequence-based reagent | *skn-1*-R | This paper | PCR primers | GAGACCCATTGGACGGTTGA |
| Sequence-based reagent | *gst-4*-F | This paper | PCR primers | TGCTCAATGTGCCTTACGAG |
| Sequence-based reagent | *gst-4*-R | This paper | PCR primers | AGTTTTTCCAGCGAGTCCAA |
| Sequence-based reagent | *gst-6*-F | This paper | PCR primers | TTTGGCAGTTGTTGAGGAG |
| Sequence-based reagent | *gst-6*-R | This paper | PCR primers | TGGGTAATCTGGACGGTTTG |
| Sequence-based reagent | *gst-7*-F | This paper | PCR primers | AGGACAACAGAATCCCAAAGG |
| Sequence-based reagent | *gst-7*-R | This paper | PCR primers | AGCAAATCCCATCTTCACCAT |
| Sequence-based reagent | *gst-10*-F | This paper | PCR primers | GTCTACCACGTTTTGGATGC |
| Sequence-based reagent | *gst-10*-R | This paper | PCR primers | ACTTTGTCGGCCTTTCTCTT |
| Sequence-based reagent | *gcs-1*-F | This paper | PCR primers | AATCGATTCCTTTGGAGACC |
| Sequence-based reagent | *gcs-1*-R | This paper | PCR primers | ATGTTTGCCTCGACAATGTT |
| Sequence-based reagent | *ctl-1*-F | This paper | PCR primers | GCGGATACCGTACTCGTGAT |
| Sequence-based reagent | *ctl-1*-R | This paper | PCR primers | GTGGCTGCTCGTAGTTGTGA |
| Sequence-based reagent | *prdx-3*-F | This paper | PCR primers | CTTGACTTCACCTTTGTATGCC |
| Sequence-based reagent | *prdx-3*-R | This paper | PCR primers | GGCGATCTTCTTGTTGAAATCA |
| Sequence-based reagent | *mtl-1*-F | This paper | PCR primers | CAAGTGTGACTGCAAAAACAAG |
| Sequence-based reagent | *mtl-1*-R | This paper | PCR primers | GCAGTACTTCTCACAACACTTG |

*Continued*

| Reagent type (species) or resource | Designation | Source or reference | Identifiers | Additional information |
|---|---|---|---|---|
| Sequence-based reagent | *osm-9*-F | This paper | PCR primers | TTCGGTTGGATCAGGAAGGC |
| Sequence-based reagent | *osm-9*-R | This paper | PCR primers | GCTTGCTTTCTCTGACGTGC |
| Sequence-based reagent | *ocr-2*-F | This paper | PCR primers | ACTTGTAGATATGCATGGCGGT |
| Sequence-based reagent | *ocr-2*-R | This paper | PCR primers | CCAAGTCGTTCATTTCTTTCCTTA |

## Strains

*C. elegans* were kept at 20°C on the nematode growth media (NGM) with plates seeded with *E. coli* OP50. Strains used in this study were obtained from *Caenorhabditis* Genetics Center. The *C. elegans* strains used in this study were as follows: wild-type Bristol strain *N2*, CX4544 *ocr-2(ak47) IV*, CX10 *osm-9(ky10) IV*, AM140 rmIs132 [unc-54p::Q35::YFP], LG333 *skn-1(zu135) (IV)/nT1[qIs51] (IV;V)*; *ldIs7 [skn-1b/c::gfp]*, CF1038 *daf-16(mu86) (I)*, CL2166 dvIs19 *[gst-4p::gfp::NLS] (III)*, EU31 *skn-1(zu135) (IV)/ nT1[unc-?(n754);let-?] (IV;V)*.

## Lifespan analysis

Worms were synchronized with bleaching buffer, followed by the starvation in M9 buffer at the L1 stage for 24 hr. Worms were thereafter transferred to the NGM plates containing the respective compounds at L4 stage. To avoid progeny hatching, 50 µg/mL of FUdR was added to the agar plates from day 0 to day 10. From the 10th day of adulthood, all the groups were transferred to the plates without compounds or FUdR treatment until the end of life. During adulthood, worms were counted every day and transferred to the fresh plates every 2 days. Death was indicated by total cessation of movement in response to gentle mechanical stimulation. The survival curves were generated using GraphPad Prism 8.3.0. The log-rank (Mantel-Cox) test was used to assess the curve significance.

For lifespan screening experiments, 15 worms were cultured on each Petri dish (60 mm in diameter) containing NGM plate (*Petrascheck et al., 2007*; *Evason et al., 2005*). 1 Petri dishes in the 1st round screen, 2 Petri dishes in the 2nd round screen, 4 Petri dishes in the 3rd round screen and 8 Petri dishes in the validation of the effect of drugs were used. In lifespan assay of mutant or control worms, more than 100 worms were used for one experiment and a least three independent experiments were performed for biological replication.

## RNAi experiment

*E. coli* strain HT115 was used for this assay. The clones used were *osm-9* (B0212.5) and *ocr-2* (T09A12.3). L4440 was used as the vector. Worms fed with the bacteria expressing L4440 or engineered to produce a gene RNAi effect were cultured until the F4 stage. One subset of the worms was confirmed to exhibit the decreased expression of the gene via qPCR. Then, the other subset of worms was synchronized for a lifespan assay on the control and JM03 400 µM treated NGM plates seeded with bacteria either expressing L4440 or engineered to produce a gene RNAi effect.

## Bacterial growth assay

A single colony of bacteria was inoculated in the LB media and cultured at 37 °C. For plate assay, 30 µL of bacterial culture ($OD_{600}$ = 0.4) was transferred to an NGM plate either with or without crotamiton or JM03 at a concentration of 400 µM, and cultured at 20 °C. The bacteria were washed off using 1 mL M9 buffer and $OD_{600}$ was measured every 12 hr, with M9 buffer as the blank control. OD was assessed using a Hitachi U-2910 spectrometer with a 10 mm quartz cuvette. At least three technical and three biological independent replicates were performed.

## Thrashing assay

Wild-type worms *N2* were transferred to the NGM plates at L4 stage and incubated with JM03 at the concentration of 400 µM. For the control and JM03 treatment groups, thrashes were counted on days 3, 8, and 12. Any change in the midbody bending direction was referred to as a thrash. Worms were placed in M9 buffer drop on an NGM plate without OP50 and allowed to adapt for 30 s. Then, the number of thrashes over 30 s were counted.

In thrashing assay, pharyngeal pumping assay and reproductive lifespan assay, more than 15 worms were used for one experiment and a least three independent experiments were performed for biological replication.

## Pharyngeal pumping assay

Wild-type worms *N2* were transferred to the NGM plates at L4 stage with JM03. On days 3, 6, 9, and 12, worms were evaluated by quantifying the contractions of the pharynx over a period of 30 s.

## Reproductive lifespan assay

Wild-type worms *N2* were transferred to 3.5 cm NGM plates individually at L4 stage with or without JM03 at a concentration of 400 µM. They were further moved to a fresh plate each day until 3 consecutive days without the progeny production. After transferring, plates were checked for progeny after 2 days. For each individual, the last day of the live progeny production was determined as the day of reproductive cessation.

## Cell culture and viability assay

MRC-5 cells were maintained in MEM medium (Gibco) supplemented with 10% FBS (42F6590K, Gibco), 1% non-essential amino acid solution (BI), 1% sodium pyruvate solution (BI), and 1% Penicillin-Streptomycin (100×) (Yeasen). Thereafter, the cells were cultured at 37 °C in an incubator with humidified atmosphere and with 5% $CO_2$. Cells were periodically checked to be mycoplasma-free using GMyc-PCR Mycoplasma Test Kit (40,601ES10, Yeasen, Shanghai, China).

For the cell counting kit-8 (CCK-8) assay, MRC-5 cells (1 × $10^4$ cells/well) were seeded in the 96-well culture plates (100 µL/well) for 12 hr. Further, different concentrations of test compounds were added to the plates in 100 µL of fresh medium (the total volume was 200 µL, DMSO <0.1%) and incubated for 72 hr. After removal of the cell culture medium, a 10% CCK-8 solution (Targetmol, C0005) in medium was added and re-incubated for 2 hr. Then, the absorbance at 450 nm was measured in a microplate reader (Biotek, Vermont, USA). At least three technical and three biological independent replicates were performed.

## Hypertonic stress resistance assay

Worms were transferred to the NGM plates at L4 stage and incubated with JM03 at a concentration of 400 µM for 4 days. Approximately 60 worms were transferred to NGM plates with 500 mM NaCl and their movement (# moving/total) was assessed. Worms were defined as paralyzed if they failed to move forward upon the tail prodding. Survival was measured every 2 min until all the worms were paralyzed. More than 100 worms were used for one experiment and a least three independent experiments were performed for biological replication.

## Oxidative stress resistance assay

Worms were transferred to the NGM plates at L4 stage and incubated with JM03 at a concentration of 400 µM for 4 days. Worms were further transferred to the 24-well plate (6 worms/well) and incubated in M9 buffer containing 10 mM paraquat. Survival was measured every two hours until all the worms were died. More than 70 worms were used for one experiment and a least three independent experiments were performed for biological replication.

## PolyQ aggregation assay

PolyQ aggregation assay was performed using AM140 *C. elegans* expressing polyQ35::YFP fusion protein in muscle cells. Worms were transferred to the NGM plates at L4 stage and incubated with JM03 at a concentration of 400 µM for 4 days. Thereafter, YFP images were taken using a fluorescence microscopy (Nikon Eclipse Ti2) and the polyQ35::YFP aggregates in worms were quantified manually using imageJ software. More than 50 worms were used for one experiment and a least three independent experiments were performed for biological replication.

## Transcriptome analysis by RNA-sequencing

The transcriptome analysis by RNA sequencing was performed according to a previously published method (*Bao et al., 2021*). Wild type worms *N2* were transferred to the NGM plates at the L4 stage

and incubated with JM03 at a concentration of 400 μM for 10 days. At the 10th day of adulthood, worms were collected. Total RNAs were extracted using Trizol Reagent (R0016, Beyotime, Shanghai, China). Further assay and analysis were assisted by Majorbio Bio-Pharm Technology Co. Ltd (Shanghai, China). The accession number for RNA-seq reported in this paper is GEO: GSE200459.

## qRT-PCR analysis

For the qRT-PCR analysis of MRC-5 cells, cells were exposed to JM03 at a concentration of 400 μM for 3 days. For the qRT-PCR analysis of *C. elegans*, wild type worms *N2* were transferred to the NGM plates at the L4 stage and incubated with JM03 at a concentration of 400 μM for 4 days. Total RNA was extracted from *C. elegans* or cells with a Total RNA Kit II (R6934-01, Omega, USA) and reverse transcribed using Hifair Ⅱ 1st Strand cDNA Synthesis SuperMix for qPCR (11,123ES60, Yeasen, Shanghai, China) in accordance with the manufacturer's instructions. qPCR was performed using Hieff qPCR SYBR Green Master Mix (11,201ES08, Yeasen, Shanghai, China) on a CFX96 quantitative PCR system (Bio-Rad, USA). Data were processed using CFX Maestro 1.0.

## SKN-1 and GST-4 expression determination

CL2166 (*gst-4 p::gfp*) and LG333 (*skn-1b/c::gfp*) transgenic strains were transferred to the NGM plates at L4 stage and incubated with 400 μM JM03 for 4 days. The SKN-1 and GST-4 expression was determined by the fluorescence microscopy (Nikon Eclipse Ti2). The fluorescence intensity was quantified using the ImageJ software. More than 50 worms were used for 1 experiment and a least three independent experiments were performed for biological replication.

## Statistical analysis

All the data are represented as mean ± SD. Statistical analysis was conducted using Graphpad Prism 8.3.0 and significant differences within treatments were determined by Log-rank (Mantel-Cox) test, two-way ANOVA or Student's t-test. $p \leq 0.05$ was considered statistically significant.

## General information of crotamiton derivatives

All the reagents were purchased from commercial corporation without further purification. Nuclear magnetic resonance (NMR) spectroscopy was recorded on 400 MHz or 600 MHz Bruker spectrometer at 303 K and referenced to TMS. Chemical shifts were reported in parts per million (ppm, $\delta$). High-resolution mass spectra (HRMS) data were given by Waters LCT or Agilent 6,545 Q-TOF. The flash column chromatography was conducted on silica gel (200–300 mesh) and visualized under UV light at 254 and 365 nm.

*General procedures for the synthesis of **JM01-JM05, JM10, JM12, JM13, JM15, 9–11***

To a solution of **1–8** (4.0 mmol) in dichloromethane was added acryloyl chloride derivatives (4.0 mmol) and potassium carbonate (1.66 g, 12.0 mmol) at 0 °C. Then the mixture was stirred at room temperature for about 1 h. After removing the solvent under reduced pressure, the residue was dissolved in ethyl acetate, washed with water and brine. Then the organic phase was dried with sodium sulfate and concentrated *in vacuo*. The crude compound was purified by silica gel column chromatography.

*(E)-N-ethyl-N-(m-tolyl)but-2-enamide* (**JM01**)

780 mg, 96.0% yield.**¹H NMR** (600 MHz, CDCl₃) $\delta$ 7.33–7.27 (m, 1 H), 7.16 (d, $J$ = 7.5 Hz, 1 H), 6.97 (s, 1 H), 6.96–6.87 (m, 2 H), 5.68 (d, $J$ = 14.8 Hz, 1 H), 3.80 (q, $J$ = 6.9 Hz, 2 H), 2.39 (s, 3 H), 1.72 (d, $J$ = 6.7 Hz, 3 H), 1.13 (t, $J$ = 7.0 Hz, 3 H). ESI-HRMS [M + H]⁺ calcd for $C_{13}H_{18}NO$: 204.1383, found: 204.1389.

*(E)-N-ethyl-N-(p-tolyl)but-2-enamide* (**JM02**)

789 mg, 97.0% yield.**¹H NMR** (600 MHz, CDCl₃) $\delta$ 7.21 (d, $J$ = 7.5 Hz, 2 H), 7.03 (d, $J$ = 7.4 Hz, 2 H), 6.94–6.86 (m, 1 H), 5.68 (d, $J$ = 15.0 Hz, 1 H), 3.79 (q, $J$ = 7.0 Hz, 2 H), 2.39 (s, 3 H), 1.71 (d, $J$ = 6.7 Hz, 3 H), 1.12 (t, $J$ = 7.0 Hz, 3 H). ESI-HRMS [M + H]⁺ calcd for $C_{13}H_{18}NO$: 204.1383, found: 204.1386.

*(E)-N-ethyl-N-(2-fluorophenyl)but-2-enamide* (**JM03**)

791 mg, 95.4% yield.**¹H NMR** (600 MHz, CDCl₃) $\delta$ 7.39–7.34 (m, 1 H), 7.23–7.16 (m, 3 H), 6.99–6.91 (m, 1 H), 5.63 (d, $J$ = 15.0 Hz, 1 H), 3.87–3.81 (m, 1 H), 3.78–3.69 (m, 1 H), 1.73 (dd, $J$ = 6.9, 1.5 Hz, 3 H), 1.12 (t, $J$ = 7.2 Hz, 3 H). ESI-HRMS [M + H]⁺ calcd for $C_{12}H_{15}FNO$: 208.1132, found: 208.1135.

*(E)-N-(2-chlorophenyl)-N-ethylbut-2-enamide* (**JM04**)

828 mg, 92.5% yield. **¹H NMR** (600 MHz, CDCl₃) δ 7.54–7.50 (m, 1 H), 7.37–7.32 (m, 2 H), 7.24–7.20 (m, 1 H), 6.97–6.91 (dq, J = 13.9, 6.9 Hz, 1 H), 5.53–5.46 (m, 1 H), 4.07 (dq, J = 14.3, 7.2 Hz, 1 H), 3.48 (dq, J = 14.3, 7.2 Hz, 1 H), 1.72 (dd, J = 6.9, 1.6 Hz, 3 H), 1.14 (t, J = 7.2 Hz, 3 H). ESI-HRMS [M + H]⁺ calcd for C₁₂H₁₅ClNO: 224.0837, found: 224.0804.

*(E)-N-(2-bromophenyl)-N-ethylbut-2-enamide* (**JM05**)

1009 mg, 94.1% yield. **¹H NMR** (600 MHz, CDCl₃) δ 7.70 (dd, J = 8.0, 1.3 Hz, 1 H), 7.39 (td, J = 7.6, 1.3 Hz, 1 H), 7.28–7.25 (m, 1 H), 7.22 (dd, J = 7.8, 1.6 Hz, 1 H), 6.95 (dq, J = 13.9, 6.9 Hz, 1 H), 5.48 (dd, J = 15.0, 1.6 Hz, 1 H), 4.14 (dq, J = 14.3, 7.2 Hz, 1 H), 3.39 (dq, J = 14.2, 7.2 Hz, 1 H), 1.72 (dd, J = 6.9, 1.6 Hz, 3 H), 1.15 (t, J = 7.2 Hz, 3 H). ESI-HRMS [M + H]⁺ calcd for C₁₂H₁₅BrNO: 268.0332, found: 268.0308.

*Ethyl (E)–4-(ethyl(p-tolyl)amino)–4-oxobut-2-enoate* (**JM10**)

951 mg, 91.0% yield. **¹H NMR** (600 MHz, CDCl₃) δ 7.22 (d, J = 8.0 Hz, 2 H), 7.02 (d, J = 8.1 Hz, 2 H), 6.82 (q, J = 15.3 Hz, 2 H), 4.16 (q, J = 7.1 Hz, 2 H), 3.83 (q, J = 7.1 Hz, 2 H), 2.39 (s, 3 H), 1.24 (t, J = 7.1 Hz, 3 H), 1.15 (t, J = 7.1 Hz, 3 H). **¹³ C NMR** (150 MHz, CDCl₃) δ 165.81, 163.59, 138.29, 138.26, 134.83, 130.71, 130.42, 127.84, 60.92, 44.66, 21.10, 14.10, 12.81. ESI-HRMS [M + H]⁺ calcd for C₁₅H₂₀NO₃: 262.1438, found: 262.1435.

*Ethyl (E)–4-(ethyl(2-fluorophenyl)amino)–4-oxobut-2-enoate* (**JM12**)

960 mg, 90.5% yield. **¹H NMR** (600 MHz, CDCl₃) δ 7.43–7.37 (m, 1 H), 7.25–7.17 (m, 3 H), 6.87 (d, J = 15.2 Hz, 1 H), 6.74 (d, J = 15.2 Hz, 1 H), 4.16 (q, J = 7.1 Hz, 2 H), 3.90–3.77 (m, 2 H), 1.25 (t, J = 7.1 Hz, 3 H), 1.15 (t, J = 7.1 Hz, 3 H). ESI-HRMS [M + H]⁺ calcd for C₁₄H₁₇FNO₃: 266.1187, found: 266.1185.

*(E)-N-ethyl-N-(4-fluorophenyl)but-2-enamide* (**JM13**)

792 mg, 95.5% yield. **¹H NMR** (600 MHz, CDCl3) δ 7.15–7.09 (m, 4 H), 6.92 (dq, J = 14.0, 6.8 Hz, 1 H), 5.62 (d, J = 14.9 Hz, 1 H), 3.78 (q, J = 7.1 Hz, 2 H), 1.73 (d, J = 6.8 Hz, 3 H), 1.13 (t, J = 7.1 Hz, 3 H). ESI-HRMS [M + H]⁺ calcd for C₁₂H₁₅FNO: 208.1132, found: 208.1132.

*Ethyl (E)–4-(ethyl(4-fluorophenyl)amino)–4-oxobut-2-enoate* (**JM15**)

992 mg, 93.5% yield. **¹H NMR** (600 MHz, CDCl₃) δ 7.13 (d, J = 6.4 Hz, 4 H), 6.84 (d, J = 15.3 Hz, 1 H), 6.74 (d, J = 15.3 Hz, 1 H), 4.17 (q, J = 7.1 Hz, 2 H), 3.83 (q, J = 7.1 Hz, 2 H), 1.25 (t, J = 7.1 Hz, 3 H), 1.15 (t, J = 7.1 Hz, 3 H). ESI-HRMS [M + Na]⁺ calcd for C₁₄H₁₆FNO₃Na: 288.1006, found: 288.1010.

*General procedures for the synthesis of* **JM06-JM09, JM11, JM14**

To a solution of **JM10, JM12, JM15, 9–11** (2.0 mmol) in methanol (5 mL) was added 1 M NaOH (5 mL). The resulting mixture was stirred at room temperature for 2 h. Then the methanol was removed under reduced pressure, and the residue was acidified to pH = 2 or below with HCl (1 M). Then the solution was extracted with ethyl acetate and the combined organic solvents were dried with sodium sulfate and concentrated *in vacuo*. The crude compound was purified by silica gel column chromatography.

*(E)–3-(N-ethylbut-2-enamido)–4-methylbenzoic acid* (**JM06**)

485 mg, 98.0% yield. **¹H NMR** (600 MHz, CDCl₃) δ 10.91 (s, 1 H), 8.04 (dd, J = 8.0, 1.3 Hz, 1 H), 7.86 (d, J = 1.3 Hz, 1 H), 7.43 (d, J = 8.0 Hz, 1 H), 7.01 (dq, J = 13.9, 6.9 Hz, 1 H), 5.50 (dd, J = 15.0, 1.4 Hz, 1 H), 4.11 (dq, J = 14.2, 7.2 Hz, 1 H), 3.45 (dq, J = 14.2, 7.2 Hz, 1 H), 2.29 (s, 3 H), 1.72 (dd, J = 6.9, 1.2 Hz, 3 H), 1.18 (t, J = 7.2 Hz, 3 H). ESI-HRMS [M + H]⁺ calcd for C₁₄H₁₈NO₃: 248.1281, found: 248.1281.

*(E)–4-(N-ethylbut-2-enamido)–3-methylbenzoic acid* (**JM07**)

478 mg, 97.0% yield. **¹ H NMR** (400 MHz, CDCl₃) δ 8.09 (d, J = 1.3 Hz, 1 H), 8.02 (d, J = 8.0 Hz, 1 H), 7.22 (d, J = 8.1 Hz, 1 H), 7.00 (dq, J = 13.8, 6.8 Hz, 1 H), 5.49 (d, J = 14.9 Hz, 1 H), 4.09 (dq, J = 14.0, 7.1 Hz, 1 H), 3.46 (dq, J = 13.9, 7.0 Hz, 1 H), 2.28 (s, 3 H), 1.72 (d, J = 6.8 Hz, 3 H), 1.17 (t, J = 7.1 Hz, 3 H). ESI-HRMS [M + H]⁺ calcd for C₁₃H₁₈NO: 248.1281, found: 248.1281.

*(E)–4-(ethyl(o-tolyl)amino)–4-oxobut-2-enoic acid* (**JM08**)

460 mg, 98.5% yield. **¹H NMR** (600 MHz, CDCl₃) δ 7.33–7.29 (m, 2 H), 7.27–7.23 (m, 1 H), 7.07 (d, J = 7.6 Hz, 1 H), 6.83 (d, J = 15.3 Hz, 1 H), 6.65 (d, J = 15.3 Hz, 1 H), 4.14–4.06 (m, 1 H), 3.47–3.40 (m, 1 H), 2.18 (s, 3 H), 1.16 (t, J = 7.2 Hz, 3 H). ESI-HRMS [M + H]⁺ calcd for C₁₃H₁₆NO₃: 234.1125, found: 234.1122.

*(E)–4-(ethyl(p-tolyl)amino)–4-oxobut-2-enoic acid* (**JM09**)

464 mg, 99.5% yield. **¹H NMR** (600 MHz, CDCl₃) δ 7.22 (d, J = 7.7 Hz, 2 H), 7.00 (d, J = 7.6 Hz, 2 H), 6.80 (s, 2 H), 3.83 (q, J = 7.1 Hz, 2 H), 2.39 (s, 3 H), 1.14 (t, J = 7.1 Hz, 3 H). **¹³ C NMR** (150 MHz, CDCl₃)

$\delta$ 169.96, 163.42, 138.49, 138.02, 136.52, 130.48, 129.85, 127.75, 44.79, 21.12, 12.75. ESI-HRMS [M + H]$^+$ calcd for $C_{13}H_{16}NO_3^+$: 234.1125, found: 234.1129.

*(E)–4-(ethyl(2-fluorophenyl)amino)–4-oxobut-2-enoic acid* (**JM11**)

464 mg, 97.8% yield.**$^1$H NMR** (600 MHz, CDCl$_3$) $\delta$ 7.43–7.38 (m, 1 H), 7.24–7.16 (m, 3 H), 6.83 (d, $J$ = 15.2 Hz, 1 H), 6.77 (d, $J$ = 15.2 Hz, 1 H), 3.89–3.76 (m, 2 H), 1.15 (t, $J$ = 7.2 Hz, 3 H). ESI-HRMS [M + H]$^+$ calcd for $C_{12}H_{13}FNO_3$: 238.0874, found: 238.0874.

*(E)–4-(ethyl(4-fluorophenyl)amino)–4-oxobut-2-enoic acid* (**JM14**)

471 mg, 99.2% yield.**$^1$H NMR** (600 MHz, CDCl$_3$) $\delta$ 7.13 (d, $J$ = 6.2 Hz, 4 H), 6.79 (q, $J$ = 15.3 Hz, 2 H), 3.83 (q, $J$ = 6.9 Hz, 2 H), 1.15 (t, $J$ = 7.0 Hz, 3 H). ESI-HRMS [M-H]$^-$ calcd for $C_{12}H_{11}FNO_3^-$: 236.0728, found: 236.0724.

## Acknowledgements

This study was supported by the National Natural Science Foundation of China [22037002], the Program for Professor of Special Appointment [Eastern Scholar TP2018025] at Shanghai Institutions of Higher Learning, Sponsored by Natural Science Foundation of Shanghai [21ZR1416700], the Innovation Program of Shanghai Municipal Education Commission [2021-01-07-00-02-E00104], and the Chinese Special Fund for State Key Laboratory of Bioreactor Engineering [2060204].

## Additional information

### Funding

| Funder | Grant reference number | Author |
|---|---|---|
| National Natural Science Foundation of China | 22037002 | Jian Li |
| Program for Professor of Special Appointment | TP2018025 | Zelan Hu |
| Natural Science Foundation of Shanghai | 21ZR1416700 | Zelan Hu |
| Innovation Program of Shanghai Municipal Education Commission | 2021-01-07-00-02-E00104 | Jian Li |
| Chinese Special Fund for Special Key Laboratory of Bioreactor Rngineering | 2060204 | Jian Li |

The funders had no role in study design, data collection and interpretation, or the decision to submit the work for publication.

### Author contributions

Keting Bao, Investigation, Project administration, Writing – original draft; Wenwen Liu, Data curation, Investigation; Zhouzhi Song, Investigation, Validation; Jiali Feng, Investigation; Zhifan Mao, Data curation, Formal analysis; Lingyuan Bao, Validation; Tianyue Sun, Validation, Visualization; Zelan Hu, Jian Li, Funding acquisition, Project administration, Writing – review and editing

### Author ORCIDs

Keting Bao http://orcid.org/0000-0002-6919-9308
Wenwen Liu http://orcid.org/0000-0002-1374-6069
Zelan Hu http://orcid.org/0000-0001-7820-6373
Jian Li http://orcid.org/0000-0002-7521-8798

### Decision letter and Author response

Decision letter https://doi.org/10.7554/eLife.72410.sa1
Author response https://doi.org/10.7554/eLife.72410.sa2

## Additional files

### Supplementary files
• Transparent reporting form

### Data availability
Sequencing data have been deposited in GEO under accession codes GSE200459. All data generated or analysed during this study are included in the manuscript and supporting files. Source Data files have been provided for Figures 1 to 6.

The following dataset was generated:

| Author(s) | Year | Dataset title | Dataset URL | Database and Identifier |
|---|---|---|---|---|
| Bao KT, Feng JL, Liu WW, Hu ZL Li J | 2021 | Crotamiton derivative JM03 extends lifespan and improves 1 oxidative and hypertonic stress resistance in *Caenorhabditis elegans* via inhibiting OSM-9 | https://www.ncbi.nlm.nih.gov/geo/query/acc.cgi?acc=GSE200459 | NCBI Gene Expression Omnibus, GSE200459 |

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
