## [Editor Report]

The study proceeds from a large initial *C. elegans* lifespan screen, comprising over 1000 candidate drugs, to a successful application of structure-activity analysis to yield an optimized lead compound and a likely mechanistic target. In addition to reporting novel compounds affecting lifespan, this work will be of interest to other researchers working at different stages of drug discovery, repurposing and lead optimization.

---

## [Decision Letter]

**Decision letter after peer review:**

Thank you for submitting your article "Crotamiton derivative JM03 extends lifespan and improves oxidative and hypertonic stress resistance in *Caenorhabditis elegans* via inhibiting OSM-9" for consideration by *eLife*. Your article has been reviewed by 2 peer reviewers, one of whom is a member of our Board of Reviewing Editors, and the evaluation has been overseen by Carlos Isales as the Senior Editor. The reviewers have opted to remain anonymous.

Essential revisions:

1) The cohort size of 15 animals/drugs during the screen suggests that there must have been a substantial amount of false negatives. That cohort size only detects 50% of all drugs that extend lifespan by 20% and miss roughly 80% of the drugs that extend lifespan by 10%, the cut-off criteria for hits. There is nothing wrong with this strategy, but the text should clearly state that not having found the effect of a drug in a screen is not evidence for the absence of an effect.

The authors should roughly outline their power of detection in the screen and that the way they have done the experiment will have caused them to miss a substantial number of compounds. Rough estimations for their power of detection can be derived from the tables in PMID: 28713422. The authors should clarify this point as otherwise, there is the danger that absence of evidence will be interpreted as evidence for absence, and any future group that finds a compound that has been missed will be forced to explain why they found effects while Bao et al. did not.

2) Another concern is the fact that all drugs are screened at a single concentration (100uM) – and this concentration is not modified based on the compound of interest. Dose / conc optimization is only performed at the final (third) screening stage for 10 of the confirmed hits. I appreciate that at this scale this might be the only way to go – but the authors should, again, clearly discuss this limitation as several well-known lifespan extending drugs show optimal effects at concentrations that are different from the one used for screening here.

3) Similarly, the effect of the JM03 is relatively small, making it more difficult to show that there is no effect in a given strain. To show the absence of an effect in osm-9, the authors use cohorts of roughly 150 animals in each arm to ensure that they can detect a longevity effect if it is present. The source data show that they used enough animals for figure 3 (10 x15 animals) and should detect a longevity effect of JM03 if there were one in osm-9. However, all the animals combined amounts to essentially one experiment, and a second repeat of that size is necessary for a firm conclusion in the paper. There should therefore be a second experiment with a four-way comparison N2, osm-9 +/- JM03 again with a cohort of ~150 animals each to ensure that there is no effect. Given the central importance of these data for the proposed mechanism, it is essential to present data from two independent experiments to support osm-9 to be a lifespan target, one with RNAi and one with the mutant.

4) The drug screen uses FUdR as contraceptive. This, again, is expected and probably inevitable for this scale of screen – but it might be valuable to confirm the drug effect of the best hits (or of cotamitron at least) in the absence of FUdR as FUdR is known to impact lifespan and possibly proteostasis. The authors should present these data to ensure that the effect is not dependent on FUdR being resent.

5) I appreciate the attempt to add mammalian data but the tissue culture segment of the investigation seems disconnected and raises some questions. Why would cell "viability" increase over 100%? The control cells seem viable (as far as I can judge) – is there any death in the controls? Also, statistically, it is inappropriate to do several t-tests – surely an ANOVA test would be more appropriate here. The authors should revisit this aspect of the work and maybe limit the claims to stating that no toxicity was observed. Tissue culture experiments would be more valuable if there was more connection with other parts of the study – did the authors do any mechanistic work in cells – e.g. qPCR?

6) Regarding the RNA seq data analysis; The cluster analysis shown in Figure 4i suggests that there is substantial variability in the RNAseq data. However, a principal component (PCA) plot shown in the supplemental figures and demonstrating that the three control and the JM03 treated samples cluster appropriately would strengthen confidence in the RNAseq data and the conclusions the authors draw from it. This figure should be generated and referenced appropriately in the main text.

7) The authors should cite PMID24855942 because this paper shows the longevity of osm-9 and that it may be conserved in mammals.

8) Both reviewers asked for more details on the pharmacological space covered by the initial screen. The authors should add a figure like a pie chart showing an overview of which pharmacological activities were tested.

*Reviewer #1 (Recommendations for the authors):*

There are some limitations to the lifespan screen. First, the cohort size is only 15 animals, meaning that many potential hits will have been missed and some false positives would have been generated. This is not a major concern as there is a second stage, re-screening 125 "hits", which should remove most false positives. However, the authors should maybe comment clearly on the power analysis of this design.

Another concern is the fact that all drugs are screened at a single concentration (100uM) – and this concentration is not modified based on the compound of interest. Dose / conc optimization is only performed at the final (third) screening stage for 10 of the confirmed hits. I appreciate that at this scale this might be the only way to go – but the authors should, again, clearly discuss this limitation as several well-known lifespan extending drugs show optimal effects at concentrations that are different from the one used for screening here.

The drug screen also uses FUdR as contraceptive. This, again, is expected and probably inevitable for this scale of screen – but it might be valuable to confirm the drug effect of the best hits (or of cotamitron at least) in the absence of FUdR as FUdR is known to impact lifespan and possibly proteostasis.

There is a little bit of tissue culture data on human cells. I appreciate the attempt to add mammalian data but this part of the investigation seems disconnected and raises some questions. Why would cell "viability" increase over 100%? The control cells seem viable (as far as I can judge) – is there any death in the controls? Also, statistically, it is inappropriate to do several t-tests – surely an ANOVA test would be more appropriate here. The authors should revisit this aspect of the work and maybe limit the claims to stating that no toxicity was observed. Tissue culture experiments would be more valuable if there was more connection with other parts of the study – did the authors do any mechanistic work in cells – e.g. qPCR?

The authors then carry out chemical lead optimization and try to establish a mechanism of action. Beyond some questions regarding the power of some of these studies (see minor comments), I think the structural optimization is convincing and a strong point of this manuscript – although I cannot comment on the chemistry here.

In terms of the mode of action, the evidence remains somewhat circumstantial. Yes, JM03 does not seem to further extend the lifespan of (long-lived) osm-9 KD animals -but that does not prove that it directly binds OSM9. Some further support would be useful here. For example, the authors might carry out RNAseq comparing osm-9 KD and WT with and without JM03 treatment. Of course, direct confirmation of binding would be ideal but is also hard.

*Reviewer #2 (Recommendations for the authors):*

Sample sizes:

The authors should roughly outline their power of detection in the screen and that the way they have done the experiment will have caused them to miss a substantial number of compounds.

Rough estimations for their power of detection can be derived from the tables in PMID: 28713422. As outlined in the public review section, a strategy with false negatives is fine, but the authors should clarify that. Otherwise, there is the danger that absence of evidence will turn into evidence for absence, and any future group that finds a compound that has been missed will be forced to explain why they found effects while Bao et al. did not.

Pharmacological space covered:

The authors should add a figure like a pie chart showing an overview of which pharmacological activities were tested.

Repeat osm-9:

-There should be a second experiment with a four-way comparison N2, osm-9 +/- JM03 again with a cohort of ~150 animals each to ensure that there is no effect. I know that this is a lot of work, but as is, and unless I missed it, two experiments support osm-9 to be a lifespan target, one with RNAi and one with the mutant.

RNAseq:

The cluster analysis shown in Figure 4i suggests that there is substantial variability in the RNAseq data. However, a principal component (PCA) plot shown in the supplemental figures shows that the three control and 3 JM03 treated samples cluster would strengthen confidence in the RNAseq data and the conclusions the authors draw from it.

Direct interaction:

If the authors could come up with an experiment that clearly shows that JM03 interacts with osm-9, or at least TRPV4, or reduces its activity in vitro, that would go a long way and make this a solid mechanistic paper. However, I won't insist on this since som-9 is a channel in the membrane, and it is notoriously difficult to show interactions for labs that are not specialized in channels.

Citations:

I think the authors should cite PMID24855942 because this paper shows the longevity of osm-9 and that it may be conserved in mammals.

I am hesitant to make comments on the English not being a native speaker myself, but I think there are a few mistakes still:

Line 159: should say "it was shown"

---

## [Author Response]

Essential revisions:1) The cohort size of 15 animals/drugs during the screen suggests that there must have been a substantial amount of false negatives. That cohort size only detects 50% of all drugs that extend lifespan by 20% and miss roughly 80% of the drugs that extend lifespan by 10%, the cut-off criteria for hits. There is nothing wrong with this strategy, but the text should clearly state that not having found the effect of a drug in a screen is not evidence for the absence of an effect.The authors should roughly outline their power of detection in the screen and that the way they have done the experiment will have caused them to miss a substantial number of compounds. Rough estimations for their power of detection can be derived from the tables in PMID: 28713422. The authors should clarify this point as otherwise, there is the danger that absence of evidence will be interpreted as evidence for absence, and any future group that finds a compound that has been missed will be forced to explain why they found effects while Bao et al. did not.

Thank you for your professional suggestion. We conducted a large-scale anti-aging drug screening with the sample size of 15 animals per drug at a single concentration in the first round and only the cohorts with +10% or over increase in lifespan made them the second round. The power of detection in the first-round screen is roughly less than 15% based on the tables in PMID: 28713422. As the reviewer’s suggested, we roughly outlined the power of detection in our screening and stated that negative compounds might be false negatives in the manuscript. So, it is probable to discover lifespan-extension ability of some negative compounds in the future study. Please see page 12 line 247-255, page 13 line 258-260 in the revised manuscript.

2) Another concern is the fact that all drugs are screened at a single concentration (100uM) – and this concentration is not modified based on the compound of interest. Dose / conc optimization is only performed at the final (third) screening stage for 10 of the confirmed hits. I appreciate that at this scale this might be the only way to go – but the authors should, again, clearly discuss this limitation as several well-known lifespan extending drugs show optimal effects at concentrations that are different from the one used for screening here.

We thank the reviewer for this helpful suggestion. All drugs were screened in the first and second round at a single concentration (100 μM) and this concentration is not modified based on the compound of interest. The concentration optimization was only performed in the third-round screen for the confirmed 10 hit compounds. In this process, we missed several positive compounds because of concentration setting deficiency. Following the reviewer’s suggestion, we added detailed discussion of this limitation. Please see page 13 line 255-260 in the revised manuscript.

3) Similarly, the effect of the JM03 is relatively small, making it more difficult to show that there is no effect in a given strain. To show the absence of an effect in osm-9, the authors use cohorts of roughly 150 animals in each arm to ensure that they can detect a longevity effect if it is present. The source data show that they used enough animals for figure 3 (10 x15 animals) and should detect a longevity effect of JM03 if there were one in osm-9. However, all the animals combined amounts to essentially one experiment, and a second repeat of that size is necessary for a firm conclusion in the paper. There should therefore be a second experiment with a four-way comparison N2, osm-9 +/- JM03 again with a cohort of ~150 animals each to ensure that there is no effect. Given the central importance of these data for the proposed mechanism, it is essential to present data from two independent experiments to support osm-9 to be a lifespan target, one with RNAi and one with the mutant.

We thank the reviewer for this professional suggestion. As the reviewer suggested, we have conducted the other two independent experiments to verify OSM-9 to be a longevity target, one with *osm-9* RNAi and one with the *osm-9* mutant. Unfortunately, lifespan experiments with *osm-9* RNAi were interrupted at day 13 (Author response image 1) and those with *osm-9* mutant were interrupted at day 9 by the sudden outbreak of coronavirus epidemic in the university. All the graduate students have not been allowed to enter the labs since 1st, March 2022 because of the severe situation of COVID-19 epidemic in Shanghai.

**Author response image 1. sa2fig1:** lifespan assay of *osm-9* RNAi worms (a-c) or *osm-9* mutant worms (d). Data are compared using the Log-rank test. * *P* < 0.05, *** *P* < 0.001, **** *P* < 0.0001. N.S., not significant. The resource data of the figures is also provided.

We apologize that we did not complete our experiments with a cohort of ~150 animals. But we have completed the lifespan experiments of *osm-9* RNAi with a cohort of ~120 animals (Author response image 1) and a cohort of ~90 animals (Author response image 1). We have also completed the lifespan experiments of *osm-9* mutant with a cohort of ~70 animals (Author response image 1). These results are consistent with Figure 3a and Figure 3c in the revised manuscript, showing JM03 failed to extend the lifespan of *osm-9* RNAi worms or *osm-9* mutant worms. In addition, RNA-seq results showed JM03 can upregulate the expression of the genes that play known or presumptive roles in proteostasis in *N2* worms but not in *osm-9* mutant. Taken together, the lifespan results and transcriptome analysis both support OSM-9 to be a lifespan target of JM03.

If the reviewers still think we need restart and complete our experiments with a cohort of ~150 animals. we will be most grateful if you could extend the deadline, preferably, for three months or more, because we still don’t know when we can return our lab and begin to start experiments.

4) The drug screen uses FUdR as contraceptive. This, again, is expected and probably inevitable for this scale of screen – but it might be valuable to confirm the drug effect of the best hits (or of cotamitron at least) in the absence of FUdR as FUdR is known to impact lifespan and possibly proteostasis. The authors should present these data to ensure that the effect is not dependent on FUdR being resent.

Thank you for your useful suggestion. We added a lifespan assay of Cro and JM03 without FUdR and found that the two compounds could also extend lifespan in *C. elegans* (Cro, *P*=0.0513; JM03, *P*<0.0001, Figure 2c). The results showed that lifespan-extension effect of Cro and JM03 is not dependent on the existence of FUdR Please see page 7 line 132-136 in the revised manuscript.

5) I appreciate the attempt to add mammalian data but the tissue culture segment of the investigation seems disconnected and raises some questions. Why would cell "viability" increase over 100%? The control cells seem viable (as far as I can judge) – is there any death in the controls? Also, statistically, it is inappropriate to do several t-tests – surely an ANOVA test would be more appropriate here. The authors should revisit this aspect of the work and maybe limit the claims to stating that no toxicity was observed. Tissue culture experiments would be more valuable if there was more connection with other parts of the study – did the authors do any mechanistic work in cells – e.g. qPCR?

We thank the reviewer for this professional suggestion. In our manuscript, significant differences were calculated using t-tests in cell viability assays. Following the reviewer’s suggestion, we changed t-tests to ANOVA test (page 34, line 686-687; page 42, line 791-792). Here, cell viability assay was usually used to test the toxicity of chemicals under different concentrations. JM03 showed cell viability increase over 100% in 200 μM but we didn’t observe obvious cell death through light field microscopy. We really agree that it’s better to limit the claims stating that no toxicity was observed. As showed in Figure 5a in the revised manuscript, both Cro and JM03 have no toxicity in MRC-5 cells under the tested concentrations (page 5, line 95-96; page 10, line 186-187).

For further mechanism study of JM03 in MRC-5 cells, we selected the concentration of 400 μM which has no toxicity and viability increasing effects to perform a qPCR assay. As showed in Figure 5b in the revised manuscript, proteostasis related genes (*Psmb1, Hsf1, Clusterin, Nrf2, Keap1, Nqo1 and Txnrd1*) were tested and the results demonstrated JM03 indeed upregulated antioxidant genes which is accord with the mechanisms in worms (page 10, line 187-190).

6) Regarding the RNA seq data analysis; The cluster analysis shown in Figure 4i suggests that there is substantial variability in the RNAseq data. However, a principal component (PCA) plot shown in the supplemental figures and demonstrating that the three control and the JM03 treated samples cluster appropriately would strengthen confidence in the RNAseq data and the conclusions the authors draw from it. This figure should be generated and referenced appropriately in the main text.

Thank you for this helpful suggestion. In order to provide more evidence to support that JM03 function via OSM-9, we carry out RNA-seq comparing osm-9 KD and WT with and without JM03 treatment. There are 3 samples in each group (N2/control, N2/JM03, *osm-9*/control and *osm-9*/JM03 groups). The figure of principal component (PCA) plot was also generated and exhibited that samples in each group are clustered and the RNA-seq data is reliable (page 10, line 205-209; Figure 5—figure supplement 1).

7) The authors should cite PMID24855942 because this paper shows the longevity of osm-9 and that it may be conserved in mammals.

We thank the reviewer for this helpful suggestion. we have cited the recommended reference as Ref 27 (page 8, line 150; page 30, line 623-624).

8) Both reviewers asked for more details on the pharmacological space covered by the initial screen. The authors should add a figure like a pie chart showing an overview of which pharmacological activities were tested.

We thank the reviewer for this professional suggestion. Because drugs used in lifespan screening distributed in a variety of pharmacological fields, a pie chart of drugs classification was added to directly show an overview of which pharmacologically active drug is being tested (Figure 1b and page 5, line 91-93).

Reviewer #1 (Recommendations for the authors):There are some limitations to the lifespan screen. First, the cohort size is only 15 animals, meaning that many potential hits will have been missed and some false positives would have been generated. This is not a major concern as there is a second stage, re-screening 125 "hits", which should remove most false positives. However, the authors should maybe comment clearly on the power analysis of this design.

Thank you for your professional suggestion. We conducted a large-scale anti-aging drug screening with the sample size of 15 animals per drug at a single concentration in the first round and only the cohorts with +10% or over increase in lifespan made them the second round. The power of detection in the first-round screen is roughly less than 15% based on the tables in PMID: 28713422. As the reviewer’s suggested, we roughly outlined the power of detection in our screening and stated that negative compounds might be false negatives in the manuscript. So, it is probable to discover lifespan-extension ability of some negative compounds in the future study. Please see page 12 line 247-255, page 13 line 258-260 in the revised manuscript.

Another concern is the fact that all drugs are screened at a single concentration (100uM) – and this concentration is not modified based on the compound of interest. Dose / conc optimization is only performed at the final (third) screening stage for 10 of the confirmed hits. I appreciate that at this scale this might be the only way to go – but the authors should, again, clearly discuss this limitation as several well-known lifespan extending drugs show optimal effects at concentrations that are different from the one used for screening here.

We thank the reviewer for this helpful suggestion. All drugs were screened in the first and second round at a single concentration (100 μM) and this concentration is not modified based on the compound of interest. The concentration optimization was only performed in the third-round screen for the confirmed 10 hit compounds. In this process, we missed several positive compounds because of concentration setting deficiency. Following the reviewer’s suggestion, we added detailed discussion of this limitation. Please see page 13 line 255-260 in the revised manuscript.

The drug screen also uses FUdR as contraceptive. This, again, is expected and probably inevitable for this scale of screen – but it might be valuable to confirm the drug effect of the best hits (or of cotamitron at least) in the absence of FUdR as FUdR is known to impact lifespan and possibly proteostasis.

Thank you for your useful suggestion. We added a lifespan assay of Cro and JM03 without FUdR and found that the two compounds could also extend lifespan in *C. elegans* (Cro, *P*=0.0513; JM03, *P*<0.0001, Figure 2c). The results showed that lifespan-extension effect of Cro and JM03 is not dependent on the existence of FUdR Please see page 7 line 132-135 in the revised manuscript.

There is a little bit of tissue culture data on human cells. I appreciate the attempt to add mammalian data but this part of the investigation seems disconnected and raises some questions. Why would cell "viability" increase over 100%? The control cells seem viable (as far as I can judge) – is there any death in the controls? Also, statistically, it is inappropriate to do several t-tests – surely an ANOVA test would be more appropriate here. The authors should revisit this aspect of the work and maybe limit the claims to stating that no toxicity was observed. Tissue culture experiments would be more valuable if there was more connection with other parts of the study – did the authors do any mechanistic work in cells – e.g. qPCR?

We thank the reviewer for this professional suggestion. In our manuscript, significant differences were calculated using t-tests in cell viability assays. Following the reviewer’s suggestion, we changed t-tests to ANOVA test (page 34, line 685-687; page 42, line 791-792). Here, cell viability assay was usually used to test the toxicity of chemicals under different concentrations. JM03 showed cell viability increase over 100% in 200 μM but we didn’t observe obvious cell death through light field microscopy. We really agree that it’s better to limit the claims stating that no toxicity was observed. As showed in Figure 5a in the revised manuscript, both Cro and JM03 have no toxicity in MRC-5 cells under the tested concentrations (page 5, line 95-96; page 10, line 186-187).

For further mechanism study of JM03 in MRC-5 cells, we selected the concentration of 400 μM which has no toxicity and viability increasing effects to perform a qPCR assay. As showed in Figure 5b in the revised manuscript, proteostasis related genes (*Psmb1, Hsf1, Clusterin, Nrf2, Keap1, Nqo1 and Txnrd1*) were tested and the results demonstrated JM03 indeed upregulated antioxidant genes which is accord with the mechanisms in worms (page 10, line 187-190).

The authors then carry out chemical lead optimization and try to establish a mechanism of action. Beyond some questions regarding the power of some of these studies (see minor comments), I think the structural optimization is convincing and a strong point of this manuscript – although I cannot comment on the chemistry here.

We thank the reviewer for this encouraging comment.

In terms of the mode of action, the evidence remains somewhat circumstantial. Yes, JM03 does not seem to further extend the lifespan of (long-lived) osm-9 KD animals -but that does not prove that it directly binds OSM9. Some further support would be useful here. For example, the authors might carry out RNAseq comparing osm-9 KD and WT with and without JM03 treatment. Of course, direct confirmation of binding would be ideal but is also hard.

Thank you for your helpful suggestion. We really agree that it would be ideal if we prove JM03 directly binds OSM9. Unfortunately, we can’t get OSM9 protein to do this experiment. In order to provide more evidence to support that JM03 function via OSM-9, we carry out RNA-seq comparing osm-9 KD and WT with and without JM03 treatment. There are 3 samples in each group (*N2*/control, *N2*/JM03, *osm-9*/control and *osm-9*/JM03 groups). The results showed that JM03 can upregulate the expression of the genes that play known or presumptive roles in proteostasis in *N2* worms but not in *osm-9(ky10)* mutant (Figure 5e, GEO accession number: GSE200459), which supported the notion that JM03 upregulates the genes associated with proteostasis through OSM-9 (page 11, line 212-218).

Reviewer #2 (Recommendations for the authors):Sample sizes:The authors should roughly outline their power of detection in the screen and that the way they have done the experiment will have caused them to miss a substantial number of compounds.Rough estimations for their power of detection can be derived from the tables in PMID: 28713422. As outlined in the public review section, a strategy with false negatives is fine, but the authors should clarify that. Otherwise, there is the danger that absence of evidence will turn into evidence for absence, and any future group that finds a compound that has been missed will be forced to explain why they found effects while Bao et al. did not.

Thank you for your professional suggestion. We conducted a large-scale anti-aging drug screening with the sample size of 15 animals per drug at a single concentration in the first round and only the cohorts with +10% or over increase in lifespan made them the second round. The power of detection in the first-round screen is roughly less than 15% based on the tables in PMID: 28713422. As the reviewer’s suggested, we roughly outlined the power of detection in our screening and stated that negative compounds might be false negatives in the manuscript. So, it is probable to discover lifespan-extension ability of some negative compounds in the future study. Please see page 12 line 247-255, page 13 line 258-260 in the revised manuscript.

Pharmacological space covered:The authors should add a figure like a pie chart showing an overview of which pharmacological activities were tested.

We thank the reviewer for this professional suggestion. Because drugs used in lifespan screening distributed in a variety of pharmacological fields, a pie chart of drugs classification was added to directly show an overview of which pharmacologically active drug is being tested (Figure 1b and page 5, line 91-93).

Repeat osm-9:-There should be a second experiment with a four-way comparison N2, osm-9 +/- JM03 again with a cohort of ~150 animals each to ensure that there is no effect. I know that this is a lot of work, but as is, and unless I missed it, two experiments support osm-9 to be a lifespan target, one with RNAi and one with the mutant.

We thank the reviewer for this professional suggestion. As the reviewer suggested, we have conducted the other two independent experiments to verify OSM-9 to be a longevity target, one with *osm-9* RNAi and one with the *osm-9* mutant. Unfortunately, lifespan experiments with *osm-9* RNAi were interrupted at day 13 (Author response image 1) and those with *osm-9* mutant were interrupted at day 9 by the sudden outbreak of coronavirus epidemic in the university. All the graduate students have not been allowed to enter the labs since 1st, March 2022 because of the severe situation of COVID-19 epidemic in Shanghai.

We apologize that we did not complete our experiments with a cohort of ~150 animals. But we have completed the lifespan experiments of *osm-9* RNAi with a cohort of ~120 animals (Author response image 1) and a cohort of ~90 animals (Author response image 1). We have also completed the lifespan experiments of *osm-9* mutant with a cohort of ~70 animals (Author response image 1). These results are consistent with Figure 3a and Figure 3c in the revised manuscript, showing JM03 failed to extend the lifespan of *osm-9* RNAi worms or *osm-9* mutant worms. In addition, RNA-seq results showed JM03 can upregulate the expression of the genes that play known or presumptive roles in proteostasis in *N2* worms but not in *osm-9* mutant. Taken together, the lifespan results and transcriptome analysis both support OSM-9 to be a lifespan target of JM03.

If the reviewers still think we need restart and complete our experiments with a cohort of ~150 animals. we will be most grateful if you could extend the deadline, preferably, for three months or more, because we still don’t know when we can return our lab and begin to start experiments.

RNAseq:The cluster analysis shown in Figure 4i suggests that there is substantial variability in the RNAseq data. However, a principal component (PCA) plot shown in the supplemental figures shows that the three control and 3 JM03 treated samples cluster would strengthen confidence in the RNAseq data and the conclusions the authors draw from it.

Thank you for this helpful suggestion. In order to provide more evidence to support that JM03 function via OSM-9, we carry out RNA-seq comparing osm-9 KD and WT with and without JM03 treatment. There are 3 samples in each group (N2/control, N2/JM03, osm-9/control and osm-9/JM03 groups). The figure of principal component (PCA) plot was also generated and exhibited that samples in each group are clustered and the RNA-seq data is reliable (page 10, line 205-209; Figure 5—figure supplement 1).

Direct interaction:If the authors could come up with an experiment that clearly shows that JM03 interacts with osm-9, or at least TRPV4, or reduces its activity in vitro, that would go a long way and make this a solid mechanistic paper. However, I won't insist on this since som-9 is a channel in the membrane, and it is notoriously difficult to show interactions for labs that are not specialized in channels.

Thank you for your professional suggestion and understanding. It is really notoriously difficult to show interactions for a traditional pharmaceutical chemistry lab that are not specialized in channels. Alternatively, we conducted RNA-seq experiments of *N2*/control, *N2*/JM03, *osm-9*/control and *osm-9*/JM03 groups. Results showed JM03 can upregulate the expression of the genes that play known or presumptive roles in proteostasis in *N2* worms but not in *osm-9(ky10)* mutant (Figure 5e, GEO accession number: GSE200459), which provide additional evidence to support that JM03 function via OSM-9 in *C. elegans* (page 11, line 212-218).

Citations:I think the authors should cite PMID24855942 because this paper shows the longevity of osm-9 and that it may be conserved in mammals.I am hesitant to make comments on the English not being a native speaker myself, but I think there are a few mistakes still:Line 159: should say "it was shown"

We thank the reviewer for this helpful suggestion. We have cited the recommended reference as Ref 27 (page 8, line 150; page 30, line 23-24) and changed to “it was shown” (page 9, line 165).